# Discovering Scaling Exponents with Physics-Informed Müntz-Szász Networks

**Gnankan Landry Regis N'guessan** [1 2 3]   **Bum Jun Kim** [4]

## Abstract

Physical systems near singularities, interfaces, and critical points exhibit power-law scaling, yet standard neural networks leave the governing exponents implicit. We introduce physics-informed Müntz-Szász Networks (MSN-PINN), a power-law basis network that treats scaling exponents as trainable parameters. The model outputs both the solution and its scaling structure. We prove identifiability, or unique recovery, and show that, under these conditions, the squared error between learned and true exponents scales as $\mathcal{O}(|\mu - \alpha|^2)$. Across experiments, MSN-PINN achieves single-exponent recovery with 1–5% error under noise and sparse sampling. It recovers corner singularity exponents for the two-dimensional Laplace equation with 0.009% error, matches the classical result of Kondrat'ev (1967), and recovers forcing-induced exponents in singular Poisson problems with 0.03% and 0.05% errors. On a 40-configuration wedge benchmark, it reaches a 100% success rate with 0.022% mean error. Constraint-aware training encodes physical requirements such as boundary condition compatibility and improves accuracy by three orders of magnitude over naive training.

## 1. Introduction

Power-law scaling appears throughout physics, including stress fields near crack tips that diverge as $r^{-1/2}$ (Anderson, 2017), turbulent energy spectra that follow $k^{-5/3}$ (Kolmogorov, 1941), and Laplace solutions at re-entrant corners that scale as $r^{2/3}$ (Kondrat'ev, 1967). These exponents encode symmetries, conservation laws, and the mathematical structure of the governing equations. In these settings, identifying them is a primary scientific goal. Fracture toughness, universality class membership, and solution regularity depend directly on these scaling exponents.

These settings motivate the need for methods that identify exponents as part of solving the governing equations. To that end, we combine Müntz-Szász Networks (MSN)—which represent solutions using power-law bases with trainable exponents—with physics-informed neural networks (PINNs) that enforce partial differential equation (PDE) residuals and boundary conditions (BCs). We refer to the resulting approach as MSN-PINN and summarize it in Figure 1.

Despite their universal approximation capabilities (Cybenko, 1989), neural networks prioritize accurate fits over explaining why a function takes its particular form. Their opaque representations limit scientific applications. PINNs encode governing equations through residual penalties and can accurately solve both forward and inverse problems, yet even PINNs hide the underlying solution structure. While verifying that a learned function satisfies the governing PDE is straightforward, extracting scaling exponents requires fragile post-hoc analysis.

The difficulty lies in the architecture. Standard multilayer perceptrons (MLPs) represent functions as sums of piecewise-linear hinges when using rectified linear unit (ReLU) activations. Approximating $x^\alpha$ for $\alpha \in (0, 1)$ to $\epsilon$-accuracy requires $N = \Omega(\epsilon^{-2})$ units (N'guessan, 2025), and the exponent hides in millions of weights. Smooth activations such as tanh or sigmoid face analogous challenges. Their bounded derivatives prevent efficient representation of unbounded power-law behavior. Even with accurate fits, the physically meaningful exponent stays hidden.

### 1.1. The MSN Approach

MSNs (N'guessan, 2025) address this gap by making scaling exponents explicit learnable parameters rather than implicit features of hidden representations. We focus on an ansatz

[1]Axiom Research Group [2]Department of Applied Mathematics and Computational Science, The Nelson Mandela African Institution of Science and Technology, Tanzania [3]African Institute for Mathematical Sciences, Research and Innovation Centre, Rwanda [4]The University of Tokyo, Japan. Correspondence to: Bum Jun Kim <bumjun.kim@weblab.t.u-tokyo.ac.jp>.

*Proceedings of the 43$^{rd}$ International Conference on Machine Learning*, Seoul, South Korea. PMLR 306, 2026. Copyright 2026 by the author(s).

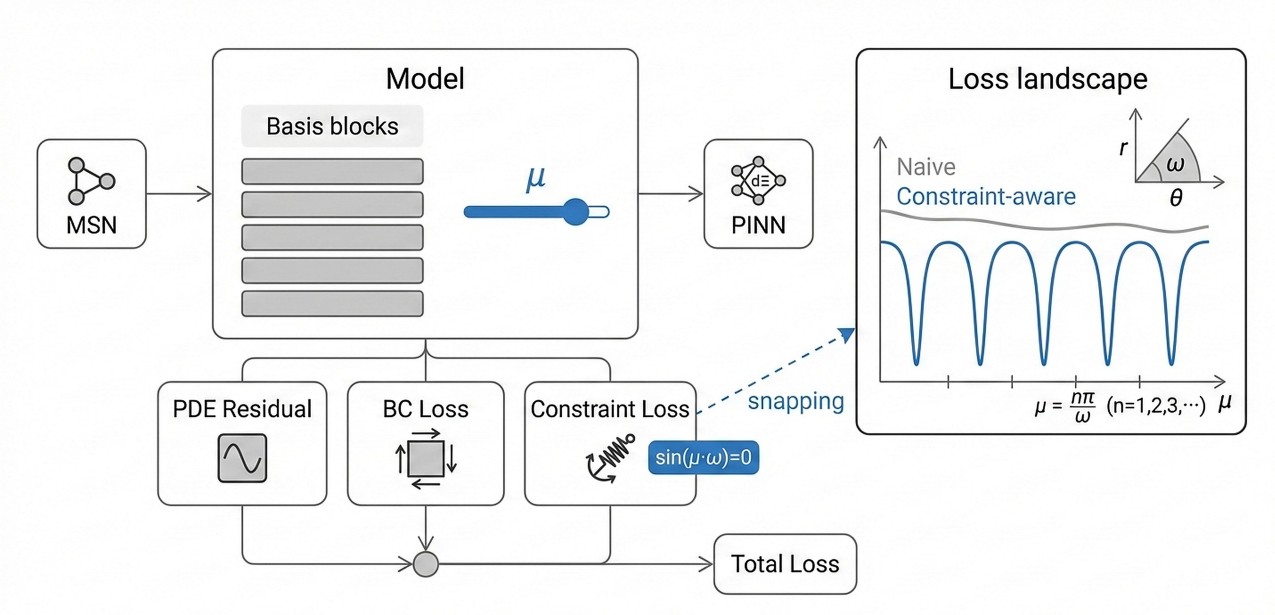

*Figure 1.* MSN-PINN discovers scaling exponents directly from physics. Unlike standard neural networks that hide exponents within millions of weights, the MSN ansatz $u(x) = \sum_k c_k x^{\mu_k}$ outputs the exponents $\{\mu_k\}$ as explicit parameters. Physics-informed losses enforce PDE residuals and BCs to guide exponent learning. For wedge singularities, constraint-aware training adds $\mathcal{L}_{\text{constraint}} = \sum_k |c_k| \sin^2(\mu_k \omega)$, which drives the exponents toward physically valid values $\mu = n\pi/\omega$—improving accuracy from 14.6% to 0.009% error.

form of MSN tailored to exponent discovery:

$$u_\theta(x) = \sum_{k=1}^{K} c_k x^{\mu_k}, \qquad (1)$$

where both the coefficients $\{c_k\}$ and the exponents $\{\mu_k\}$ are optimized during training. Unlike the millions of weights in standard neural architectures, these learned exponents are directly interpretable physical quantities. For example, when trained on data generated from $f(x) = x^{0.5}$, MSN recovers $\mu \approx 0.5$ as an explicit parameter.

The full MSN architecture introduced in prior work includes signed even and odd power bases and compositional layer stacking. Here, we specialize in a single Müntz expansion to isolate the identifiability and recovery of scaling laws. While previous studies have established MSN as a powerful approximation architecture, we address a more fundamental question about when true physical exponents can be reliably recovered from data or physics-informed PDE constraints.

## 1.2. Contributions

We make four key contributions. We embed the MSN within a physics-informed framework, enabling exponent discovery from PDE residuals without requiring labeled solution data; see Section 3. We establish when exponents can be uniquely recovered and derive stability bounds showing that recovery accuracy degrades gracefully with noise, gov-

erned by a separation condition analogous to the Rayleigh Limit; see Theorem 4.4 and Theorem 4.5. We introduce constraint-aware losses that encode physical requirements such as edge BCs, improving accuracy by three orders of magnitude over naive MSN-PINN approaches when the PDE provides insufficient gradient signals for exponent selection; see Section 3.4. We demonstrate exponent discovery across multiple settings; see Section 5. A singular ordinary differential equation (ODE) yields 1.31% error from physics alone. A corner singularity achieves 0.009% error, matching the analytical result of Kondrat'ev (1967). Singular forcing produces 0.03% and 0.05% errors on dual exponents. A 40-configuration wedge benchmark reaches a 100% success rate with 0.022% mean error; see Appendix I.

MSN combines the insights of asymptotic analysis with the scalability of neural networks, learning explicit exponents without requiring closed-form solutions. These learned exponents validate theory, suggest new physics, diagnose model misspecification, and enable extrapolation beyond the training domain.

## 2. Related Work

PINNs enforce PDE residuals to solve both forward and inverse problems (Raissi et al., 2019). Variational PINNs improve stability for stiff PDEs (Kharazmi et al., 2021), domain decomposition improves scalability (Jagtap & Kar-

niadakis, 2021), and conservative architectures encode invariants (Jagtap et al., 2020). Despite these advances, PINNs can still struggle with stiff problems (Krishnapriyan et al., 2021) and tend to exhibit spectral bias toward low-frequency modes (Rahaman et al., 2019). For singular solutions, Hu et al. (2024) embed prescribed singular forms. MSN allows the exponents to be learned.

Sparse Identification of Nonlinear Dynamics (SINDy) discovers governing equations through sparse regression over candidate libraries (Brunton et al., 2016). Extensions to PDEs apply the same idea to spatiotemporal data (Rudy et al., 2017), and latent-coordinate variants infer dynamics in hidden variables (Champion et al., 2019). Artificial Intelligence Feynman (AI Feynman) uses dimensional analysis and symmetry to search for symbolic forms (Udrescu & Tegmark, 2020). Although these methods discover discrete structure, they rely on fixed candidate exponents. MSN learns continuous exponents directly.

Operator learning methods map functions to functions: DeepONet uses branch and trunk networks for inputs and coordinates (Lu et al., 2021), while Fourier Neural Operators parameterize kernels in the spectral domain (Li et al., 2021). Kolmogorov-Arnold Networks (KANs) replace fixed activations with learnable splines (Liu et al., 2025b;a). KANs target smooth functions; MSN targets power-law singularities with explicit exponents.

Kondrat'ev (1967) characterized elliptic corner singularities, and Grisvard (2011) and Dauge (2006) further developed this theory. Numerical methods exploit this structure with graded meshes (Babuška et al., 1979), finite element methods (FEMs) such as $hp$-finite element methods (Schwab, 1998), and extended finite element methods (XFEM) for enrichment (Moës et al., 1999; Babuška & Melenk, 1997). These methods prescribe singular bases analytically. MSN learns the bases from physics.

The classical Müntz-Szász theorem (Müntz, 1914; Borwein & Erdélyi, 2012) characterizes when the spans of a set $\{x^{\lambda_n}\}$ are dense in $C[0, 1]$. Computational applications (Shen & Wang, 2016) achieve exponential convergence for singular problems. Prior work by N'guessan (2025) introduced trainable exponents; we extend to physics-informed discovery.

## 3. Method

### 3.1. The MSN Ansatz: Why Power-Law Bases?

Many physical singularities have a power-law structure. Rather than approximating $x^{0.5}$ with compositions of smooth functions, we represent it directly in a basis specifically designed for power laws.

**Definition 3.1** (MSN Ansatz)**.** An MSN with $K$ terms represents functions as

$$u_\theta(x) = \sum_{k=1}^{K} c_k x^{\mu_k}, \text{ for } x \in (0, 1], \qquad (2)$$

where $\theta = (\boldsymbol{\mu}, \boldsymbol{c})$ with exponents $\boldsymbol{\mu} = (\mu_1, \ldots, \mu_K) \in [\mu_{\min}, \mu_{\max}]^K$ and coefficients $\boldsymbol{c} = (c_1, \ldots, c_K) \in \mathbb{R}^K$.

This representation shines when approximating power-law targets:

**Proposition 3.2** (Exact Representation)**.** *If $f(x) = x^\alpha$ with $\alpha \in [\mu_{\min}, \mu_{\max}]$, then the MSN with $K = 1$ achieves zero approximation error when $\mu_1 = \alpha$ and $c_1 = 1$.*

Compared with standard networks, approximating $x^{0.5}$ with ReLU units requires $\mathcal{O}(\epsilon^{-2})$ neurons for $\epsilon$-accuracy, and the exponent 0.5 hides in the weights. MSN achieves an exact representation with one term, and the exponent becomes an explicit parameter.

### 3.2. Relationship to MSN

We continue the work on MSNs (N'guessan, 2025), which embedded learnable power-law exponents into neural architectures to obtain expressive, interpretable function approximators. Here, we specialize the MSN to its ansatz regime with a single Müntz expansion:

$$u_\theta(x) = \sum_{k=1}^{K} c_k |x|^{\mu_k}, \qquad (3)$$

with trainable exponents $\{\mu_k\}$ and linear coefficients $\{c_k\}$. This specialization isolates how MSNs learn exponents while eliminating architectural redundancies that obscure exponent identifiability.

In contrast, the full MSN architecture supports deep composition, parity-aware edge-wise even and odd representations, and can approximate general functions. We can recover the full architecture by stacking multiple Müntz expansions with inter-layer nonlinearities and signed-domain extensions. We use the reduced ansatz here because it allows us to rigorously analyze when scaling exponents can be identified, derive closed-form PDE derivatives, and optimize stably under physics-informed constraints.

MSNs learn governing scaling laws directly from data and physics. For problems requiring composition or mixed parity, the full MSN architecture can be reintroduced while retaining the identifiability results established here.

### 3.3. Physics-Informed MSN: MSN-PINN

The MSN ansatz represents the solution; physics-informed training supervises learning. MSN-PINN discovers expo-

nents from PDE constraints without requiring labeled solution data.

### 3.3.1. LOSS FUNCTION DESIGN

Consider a PDE of the form

$$\begin{aligned} \mathcal{D}[u](x) &= f(x), \text{for } x \in \Omega, \\ \mathcal{B}[u](x) &= g(x), \text{for } x \in \partial\Omega, \end{aligned} \quad (4)$$

where $\mathcal{D}$ is a differential operator and $\mathcal{B}$ encodes BCs. The MSN-PINN loss combines three components:

$$\mathcal{L}_{\text{total}}(\theta) = w_r \mathcal{L}_{\text{res}}(\theta) + w_b \mathcal{L}_{\text{BC}}(\theta) + w_s \mathcal{L}_{\text{sparse}}(\theta). \quad (5)$$

The PDE residual loss measures violation of the governing equation at the collocation points $\{x_i^r\}_{i=1}^{N_r} \subset \Omega$:

$$\mathcal{L}_{\text{res}}(\theta) = \frac{1}{N_r} \sum_{i=1}^{N_r} |\mathcal{D}[u_\theta](x_i^r) - f(x_i^r)|^2. \quad (6)$$

The BC loss enforces boundary data at points $\{x_i^b\}_{i=1}^{N_b} \subset \partial\Omega$:

$$\mathcal{L}_{\text{BC}}(\theta) = \frac{1}{N_b} \sum_{i=1}^{N_b} |\mathcal{B}[u_\theta](x_i^b) - g(x_i^b)|^2. \quad (7)$$

The sparsity loss encourages simpler solutions with fewer active terms:

$$\mathcal{L}_{\text{sparse}}(\theta) = \frac{1}{K} \sum_{k=1}^{K} |c_k|. \quad (8)$$

The sparsity term serves two purposes. First, the $\ell_1$ penalty helps interpret results by suppressing spurious terms, leaving only physically meaningful exponents with significant coefficients. Second, sparsity makes exponents easier to identify by reducing the effective model complexity.

### 3.4. Constraint-Aware Training

Naive MSN-PINN training fails on certain problems—not because the model lacks capacity, but because the physics provides an inadequate gradient signal to select exponents. Constraint-aware training adds physics-derived penalties to the loss to supply that signal.

### 3.4.1. THE EXPONENT DRIFT PROBLEM

Consider Laplace's equation on a wedge domain with an opening angle $\omega$:

$$\nabla^2 u = 0 \text{ in } \Omega, \text{ with } u|_{\theta=0} = u|_{\theta=\omega} = 0. \quad (9)$$

Solutions have the separable form $u(r, \theta) = r^\mu \Phi(\theta)$. Substituting this into Laplace's equation yields

$$\nabla^2 u = r^{\mu-2}[\mu^2 \Phi + \Phi''] = 0. \quad (10)$$

Any $\mu$ satisfies this equation provided $\Phi'' + \mu^2 \Phi = 0$, giving $\Phi(\theta) = A\sin(\mu\theta) + B\cos(\mu\theta)$.

This separation reveals the core difficulty. The Laplacian provides no gradient signal for $\mu$. Every exponent $\mu$ yields a harmonic function $r^\mu \sin(\mu\theta)$, so $\mathcal{L}_{\text{res}}$ is flat in $\mu$; all such choices achieve zero residual.

In initial experiments on the Corner Singularity problem, this degeneracy caused severe exponent drift. The network matched the boundary data with a harmonic function whose exponents were unrelated to the physical solution, producing a 14.6% error.

### 3.4.2. PHYSICAL CONSTRAINTS RESOLVE THE DEGENERACY

The resolution comes from encoding additional physical requirements. Arbitrary harmonic functions do not automatically satisfy the edge BCs $u|_{\theta=0} = u|_{\theta=\omega} = 0$. For $u = r^\mu \sin(\mu\theta)$:

$$u|_{\theta=0} = r^\mu \sin(0) = 0, \quad (11)$$

$$u|_{\theta=\omega} = r^\mu \sin(\mu\omega) = 0, \text{ so } \sin(\mu\omega) = 0. \quad (12)$$

The second condition constrains $\mu$ to a discrete set of values. $\mu \in \{\pi/\omega, 2\pi/\omega, 3\pi/\omega, \ldots\}$. For a re-entrant corner with $\omega = 3\pi/2$, the dominant exponent is $\mu = \pi/(3\pi/2) = 2/3$. We encode this constraint as an additional loss term.

$$\mathcal{L}_{\text{constraint}}(\theta) = \sum_{k=1}^{K} w_k \sin^2(\mu_k \cdot \omega), \quad (13)$$

where $w_k \propto |c_k|$ weights each term in proportion to its coefficient magnitude. Terms with larger coefficients should satisfy the constraint more precisely; terms with negligible coefficients can be ignored.

This loss term acts as a quantization force, pushing exponents toward the discrete set of physically valid values. Unlike the PDE residual, it provides a strong gradient signal for exponent correction.

$$\begin{aligned} \frac{\partial \mathcal{L}_{\text{constraint}}}{\partial \mu_k} &= w_k \cdot 2\sin(\mu_k\omega)\cos(\mu_k\omega) \cdot \omega \\ &= w_k \omega \sin(2\mu_k\omega). \end{aligned} \quad (14)$$

This gradient is nonzero whenever $\mu_k$ deviates from valid values, thereby driving convergence toward the physical solution.

### 3.4.3. GENERAL PRINCIPLE: ENCODE ALL AVAILABLE PHYSICS

The wedge example illustrates a general principle. MSN-PINN succeeds when we encode enough physics to make the exponents identifiable. Different problems need different constraints. Corner singularities use edge BCs, implying

**Algorithm 1** Constraint-Aware MSN-PINN Training

**Require:** Collocation points $\{x_i^r\}_{i=1}^{N_r}$, $\{x_i^b\}_{i=1}^{N_b}$; PDE operator $\mathcal{D}$; constraint $q(\boldsymbol{\mu})$
  Hyperparameters: $w_r, w_b, w_s, w_{\text{con}}$; $\eta_\mu \ll \eta_c$; $T$
**Ensure:** Learned exponents $\boldsymbol{\mu}^*$ and coefficients $\boldsymbol{c}^*$

INITIALIZATION
  1: $\boldsymbol{\mu} \leftarrow \text{Uniform}([\mu_{\min}, \mu_{\max}])^K$ {sample exponents}
  2: $\boldsymbol{c} \leftarrow \mathcal{N}(\mathbf{0}, 0.01\mathbf{I})$ {initialize coefficients}

TWO-TIMESCALE OPTIMIZATION
  3: **for** $t = 1$ to $T$ **do**
  4: $\quad \hat{u}(x) \leftarrow \sum_{k=1}^K c_k x^{\mu_k}$ {MSN ansatz}
  5: $\quad \mathcal{L}_{\text{con}} \leftarrow \|q(\boldsymbol{\mu})\|^2$ {constraint loss}
  6: $\quad \mathcal{L} \leftarrow w_r \mathcal{L}_{\text{res}} + w_b \mathcal{L}_{\text{BC}} + w_s \|\boldsymbol{c}\|_1 + w_{\text{con}} \mathcal{L}_{\text{con}}$
  7: $\quad \boldsymbol{c} \leftarrow \boldsymbol{c} - \eta_c \nabla_{\boldsymbol{c}} \mathcal{L}$ {fast update}
  8: $\quad \boldsymbol{\mu} \leftarrow \boldsymbol{\mu} - \eta_\mu \nabla_{\boldsymbol{\mu}} \mathcal{L}$ {slow update}
  9: **end for**
  10: **return** $(\boldsymbol{\mu}, \boldsymbol{c})$

$\sin(\mu\omega) = 0$. Crack problems use stress-free conditions, which fix angular dependence. Interface problems use continuity conditions that match exponents across interfaces. Eigenvalue problems use normalization that fixes coefficient relationships. Choosing the right constraints depends on the BCs, symmetries, and governing equations, and it makes our assumptions explicit.

### 3.5. Two-Timescale Optimization

Exponents and coefficients play different roles in MSN, requiring different optimization strategies. For fixed exponents, finding optimal coefficients reduces to a linear least-squares problem with a unique global minimum. Exponents, by contrast, appear nonlinearly and create a complex landscape with multiple local minima. This asymmetry motivates optimizing on two timescales:

$$\begin{aligned} \mu^{(t+1)} &= \mu^{(t)} - \eta_\mu \nabla_\mu \mathcal{L}, \\ c^{(t+1)} &= c^{(t)} - \eta_c \nabla_c \mathcal{L}. \end{aligned} \quad (15)$$

with $\eta_\mu < \eta_c$, typically ranging from $\eta_\mu = 0.1\eta_c$ to $\eta_\mu = 0.5\eta_c$.

The intuition is as follows. Coefficients should adapt quickly to track the current exponents, solving the inner linear problem at each step. Exponents should move slowly, exploring the outer nonlinear landscape without being destabilized by rapid coefficient changes.

Marion & Berthier (2023) prove convergence for two-timescale gradient descent, showing that the slower timescale optimizes over the manifold of solutions to the faster problem. Fast coefficient adaptation ensures that ex-

ponent gradients accurately reflect the best achievable loss for each exponent configuration. Algorithm 1 summarizes the complete procedure.

## 4. Theoretical Foundations

Three questions guide our theoretical analysis: How accurately can MSN represent power-law functions? When can exponents be uniquely recovered? How does noise affect recovery? We characterize MSN behavior near the true exponents under appropriate regularity conditions, matching the practical regime where the method operates.

### 4.1. Universal Approximation

The following result from N'guessan (2025) establishes that MSN inherits the approximation power of classical Müntz systems:

**Theorem 4.1** (Universal Approximation, (N'guessan, 2025)). *Let $f \in C[0,1]$ with $f(0) = 0$. For any $\epsilon > 0$, there exist $K \in \mathbb{N}$, exponents $\boldsymbol{\mu} \in (0, \infty)^K$, and coefficients $\boldsymbol{c} \in \mathbb{R}^K$ such that*

$$\sup_{x \in [0,1]} \left| f(x) - \sum_{k=1}^K c_k x^{\mu_k} \right| < \epsilon. \quad (16)$$

This follows directly from the Müntz-Szász theorem: The span of $\{x^{\lambda_0}, x^{\lambda_1}, \ldots\}$ with $\lambda_0 = 0$ is dense in $C[0,1]$ if and only if $\sum_{n=1}^\infty \lambda_n^{-1} = \infty$. Any sequence of positive exponents that grows sufficiently slowly satisfies this condition.

### 4.2. Approximation Rates for Power-Law Targets

For power-law targets, MSN achieves optimal rates with an explicit dependence on exponent error:

**Theorem 4.2** (Approximation Rate, (N'guessan, 2025)). *Let $f(x) = x^\alpha$ for $\alpha > 0$ and define*

$$R(\mu) = \min_{c \in \mathbb{R}} \int_0^1 (x^\alpha - cx^\mu)^2 dx. \quad (17)$$

*As $\mu \to \alpha$,*

$$R(\mu) = \mathcal{O}((\mu - \alpha)^2), \quad (18)$$

*with optimal coefficient*

$$c^*(\mu) = \frac{\int_0^1 x^{\alpha+\mu} dx}{\int_0^1 x^{2\mu} dx} = \frac{2\mu + 1}{\alpha + \mu + 1} = 1 + \mathcal{O}(|\mu - \alpha|). \quad (19)$$

We give the proof in Appendix B.

**Interpretation** The squared loss depends quadratically on the exponent error, comparing favorably with standard networks. For ReLU networks approximating $x^{0.5}$, achieving an error $\epsilon$ requires $N = \mathcal{O}(\epsilon^{-2})$ neurons; for MSN, achieving the same error requires only $|\mu - 0.5| = \mathcal{O}(\epsilon^{1/2})$. More importantly, MSN provides direct access to the exponent error, whereas a ReLU network buries the exponent within its weights.

### 4.3. Exponent Identifiability

When can exponents be uniquely recovered? We formalize identifiability as follows.

**Definition 4.3** (Exponent Identifiability). A function $f(x) = \sum_{k=1}^{K^*} c_k^* x^{\alpha_k}$ with distinct exponents $\alpha_1 < \cdots < \alpha_{K^*}$ and nonzero coefficients is identifiable if any MSN representation $\sum_{j=1}^{K} \tilde{c}_j x^{\tilde{\mu}_j}$ achieving $\|f - \sum_j \tilde{c}_j x^{\tilde{\mu}_j}\|_\infty = 0$ satisfies $K \geq K^*$ and, after reordering, $\tilde{\mu}_j = \alpha_j$ for some $j$ corresponding to the true exponent.

**Theorem 4.4** (Identifiability). *Let $f(x) = \sum_{k=1}^{K^*} c_k^* x^{\alpha_k}$ with $\alpha_1 < \cdots < \alpha_{K^*}$ and $c_k^* \neq 0$ for all $k$. Then (i) $f$ is identifiable: Any exact MSN representation must contain all the true exponents $\{\alpha_k\}$, and (ii) the representation is unique up to reordering if $K = K^*$.*

We give the proof in Appendix C.

**Interpretation** Power-law sums are uniquely determined by their exponents: The same function cannot be represented with different exponents. This makes exponent discovery possible.

### 4.4. Stability and the Separation Condition

Exact recovery requires exact data. In practice, observations are noisy, and we need stability guarantees:

**Theorem 4.5** (Stability and Separation). *Consider noisy observations $y_i = f(x_i) + \epsilon_i$ with $|\epsilon_i| \leq \sigma$, where $f(x) = \sum_{k=1}^{K^*} c_k^* x^{\alpha_k}$. Let $\hat{\mu}$ minimize the empirical loss $\hat{\mathcal{L}}(\mu, c) = \frac{1}{N} \sum_{i=1}^N (y_i - \sum_k c_k x_i^{\mu_k})^2$. If the true exponents satisfy a minimum separation condition*

$$\Delta := \min_{j \neq k} |\alpha_j - \alpha_k| > 0, \tag{20}$$

*then under appropriate conditions on the sampling points $\{x_i\}$:*

$$\min_\pi \max_k |\hat{\mu}_k - \alpha_{\pi(k)}| = \mathcal{O}(\frac{\sigma}{c_{\min}\Delta^2\sqrt{N}}), \tag{21}$$

*where $c_{\min} = \min_k |c_k^*|$ and $\pi$ ranges over all permutations.*

The proof, given in Appendix D, uses M-estimation theory.

*Table 1.* Singular ODE: Exponent discovery using only physics constraints.

| Metric | Target | Discovered |
|---|---|---|
| Dominant exponent | 0.5000 | 0.4934 |
| Relative error | — | 1.31% |
| Final PDE residual | — | $3.2 \times 10^{-6}$ |
| Final BC loss | — | $6.6 \times 10^{-4}$ |

The key insight is that the Fisher information matrix for exponent estimation, the local curvature of the loss, has a condition number depending on $\Delta^{-2}$; closer exponents yield worse conditioning.

**Remark 4.6** (On the Sampling Assumption). The sampling assumption enters through the curvature lower bound in Appendix D. Under sufficiently dense and well-spread sampling, concentration gives $\|\frac{1}{N}\sum_i \epsilon_i \phi_i\| = O(\frac{\sigma}{\sqrt{N}})$ with $(\phi_i)_k = x_i^{\alpha_k} \log x_i$. Since $\nabla_\mu \hat{\mathcal{L}}(\alpha, c^*) = -2D(\frac{1}{N}\sum_i \epsilon_i \phi_i)$ for $D = \text{diag}(c^*)$, this yields $\|\hat{\mu} - \alpha\| = O(\frac{\sigma}{c_{\min}\Delta^2\sqrt{N}})$.

### 4.5. Physics-Informed Identifiability

For PDE-constrained problems, the underlying physics provides additional structure affecting identifiability. The next proposition illustrates this with singular forcing.

**Proposition 4.7** (PDE-Induced Exponent Constraints). *Consider the Poisson equation $-u'' = x^\beta$ on $(0,1]$ with $u(0) = u(1) = 0$. If $\beta > -1$, the solution has the form*

$$u(x) = a_1 x + a_2 x^{\beta+2}, \tag{22}$$

*for specific constants $a_1, a_2$ depending on $\beta$. Thus, the forcing exponent $\beta$ induces a solution exponent $\alpha = \beta + 2$.*

We give the proof in Appendix E. This explains why MSN-PINN can discover $\alpha = 3/2$ from the singular forcing $x^{-1/2}$: The forcing exponent $\beta = -1/2$ deterministically produces the solution exponent $\alpha = \beta + 2 = 3/2$.

**Theorem 4.8** (Constraint-Aware Critical Points). *For the Laplacian on a wedge with an opening angle $\omega$, consider the augmented loss*

$$\mathcal{L}(\theta) = \mathcal{L}_{res}(\theta) + \mathcal{L}_{BC}(\theta) + \lambda \sum_k |c_k| \sin^2(\mu_k \omega). \tag{23}$$

*Any critical point $(\mu^*, c^*)$ with $\mathcal{L}_{res}(\mu^*, c^*) = 0$, $\mathcal{L}_{BC}(\mu^*, c^*) = 0$, and $|c_k^*| > 0$ for some $k$ satisfies $\sin(\mu_k^*\omega) = 0$, that is, $\mu_k^* \in \{n\pi/\omega : n \in \mathbb{N}\}$.*

We give the proof in Appendix F.

*Table 2.* Corner Singularity: Laplace wedge with constraint-aware training.

| Metric | Target | Discovered |
|---|---|---|
| Primary exponent | 0.6667 | 0.6667 |
| Relative error | — | 0.009% |
| Second harmonic | 1.3333 | 1.3335 |
| Edge constraint $\sin(\mu\omega)$ | 0 | $4.9 \times 10^{-7}$ |

**Interpretation** This theorem explains why constraint-aware training succeeds where naive training fails. Without the constraint term, the loss landscape has a continuum of global minima—all harmonic functions matching boundary data. The constraint term breaks this symmetry, isolating minima at physically valid exponents.

# 5. Experiments

We evaluate MSN-PINN in three main physics-informed settings: singular ODEs, wedge corner singularities, and singular forcing, and compare against a standard PINN with post-hoc exponent fitting in the main physics-informed settings. The primary metrics used are exponent error and constraint satisfaction.

We validate MSN-PINN for physics-informed exponent discovery: recovering exponents solely from PDE constraints, without labeled solution data, and testing constraint-aware training on challenging singular problems. Appendix I provides additional experiments characterizing robustness to noise, sampling density, identifiability limits, a comprehensive 40-configuration wedge benchmark, and a Mode-III crack example connecting the wedge formulation to fracture mechanics.

## 5.1. Singular ODE: Physics-Informed Discovery

We recover exponents from PDE constraints rather than from labeled solution data. The boundary-layer equation

$$xu''(x) + \frac{1}{2}u'(x) = 0,$$
$$u(1) = 1, \tag{24}$$

has solution $u(x) = \sqrt{x}$ with an exponent $\alpha = 0.5$. We train the MSN-PINN with $K = 4$ terms on the PDE residual loss without solution labels, concentrating collocation points near $x = 0$. Quantitative results for this setting are summarized in Table 1.

MSN-PINN discovers the exponent from physics alone. Without any labeled solution data, the network learns that $\mu \approx 0.5$ is the unique exponent consistent with the differential equation and BC. This demonstrates that MSN-PINN extracts interpretable structure directly from physical laws.

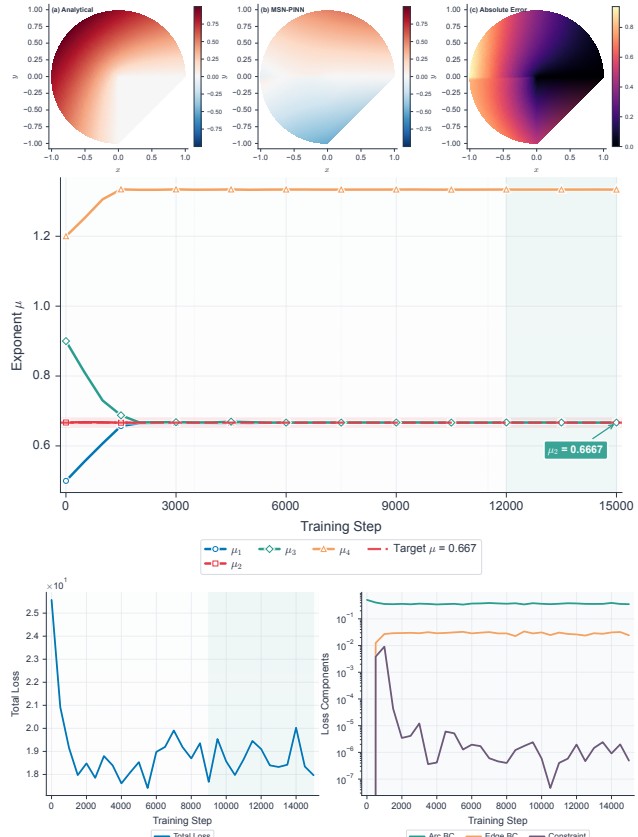

*Figure 2.* Corner singularity: MSN-PINN recovers the result of Kondrat'ev (1967) with a 0.009% error. Top: Learned solution $u(r, \theta) = r^{2/3} \sin(2\theta/3)$ on a 270° wedge domain, where color indicates solution magnitude, with blue at 0 and yellow at 1. Middle: Exponent trajectory during training—without constraints, not shown, $\mu$ drifts to 0.57 with a 14.6% error; with constraints, $\mu$ converges to the theoretical value of 0.6667. Bottom: Training loss showing rapid convergence once the constraint term activates around epoch 2000.

The exponent trajectory shows initial exploration followed by convergence: Early in training, the exponents wander as the network searches for a consistent solution; once the PDE residual begins to decrease, the exponents lock onto the correct value.

## 5.2. Corner Singularity as Main Result

The corner singularity problem demonstrates the constraint-aware MSN-PINN on a challenging 2D domain. Consider Laplace's equation defined on a wedge with an opening angle $\omega = 3\pi/2$, corresponding to a 270° re-entrant corner:

$$\nabla^2 u = 0,$$
$$u|_{\theta=0} = u|_{\theta=\omega} = 0, \tag{25}$$
$$u|_{r=1} = \sin(2\theta/3).$$

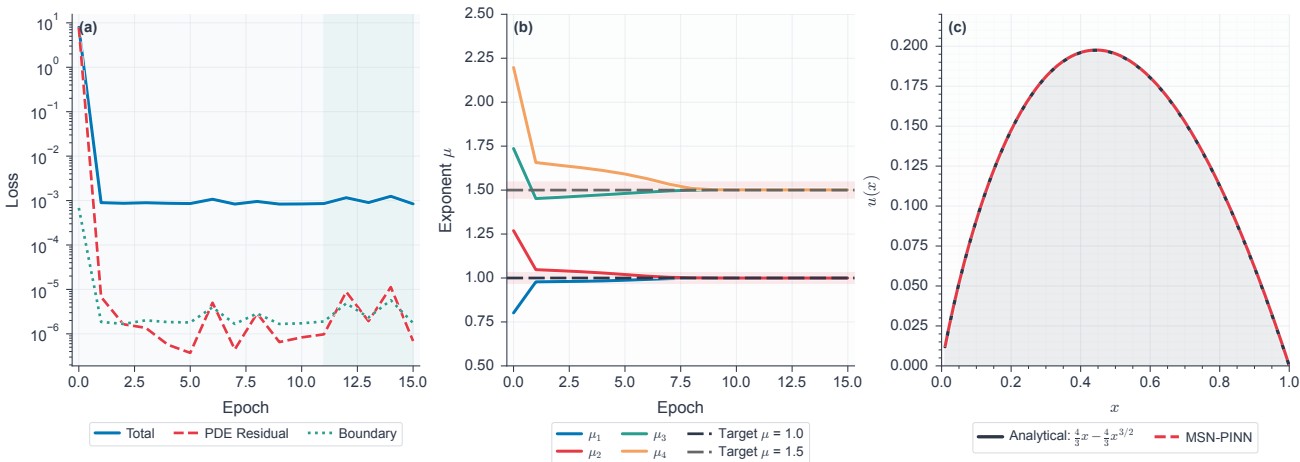

*Figure 3.* Singular forcing: MSN-PINN accurately discovers both solution exponents with <0.1% error. The Poisson equation, $-u'' = x^{-1/2}$, has the solution $u(x) = \frac{4}{3}x - \frac{4}{3}x^{3/2}$ containing exponents 1.0 for the homogeneous term and 1.5 for the particular term. Panel (a) shows the training loss convergence over 10,000 epochs. Panel (b) shows four MSN exponents initialized randomly; two converge to 1.0 with 0.03% error, and two to 1.5 with 0.05% error. The redundant convergence indicates robust discovery. Panel (c) shows the learned solution, dashed, matching the analytical solution, solid, across the domain.

*Table 3.* Singular Forcing: Dual exponent discovery.

| Target Exponent | Discovered | Error % |
|---|---|---|
| 1.0 homogeneous | 0.9997 | 0.03 |
| 1.5 particular | 1.5008 | 0.05 |

The analytical solution is $u(r, \theta) = r^{2/3} \sin(2\theta/3)$ with exponent $\alpha = 2/3 \approx 0.6667$—a classical result due to Kondrat'ev (1967).

As discussed in Section 3.4, the Laplacian provides no gradient signal for exponent selection: All functions $r^\mu \sin(\mu\theta)$ are harmonic. Without the constraint term, training on residual and boundary data converges to $\mu \approx 0.57$, giving a 14.6% error. Although the network matches the boundary data during training, it uses an incorrect exponent.

Adding $\mathcal{L}_{\text{constraint}} = \sum_k |c_k| \sin^2(\mu_k \omega)$ changes the outcome. Constraint-aware training achieves 0.009% error and recovers the analytical result of Kondrat'ev (1967) to four significant figures, representing a 1600-fold improvement over the naive approach.

This constraint uses only the compatibility family implied by the geometry and edge BCs, not the target exponent itself; Table 15 shows that small angle errors produce predictable shifts and that $\omega$ can be selected by held-out boundary validation when it is unknown.

The constraint loss creates strong gradients pushing exponents toward valid values. Unlike the PDE residual, which is flat with respect to $\mu$ for harmonic functions, the constraint term $\sin^2(\mu\omega)$ has sharp minima at $\mu = n\pi/\omega$. Combined with boundary data that selects the $n = 1$ mode, the opti-

mization landscape has a unique, well-defined minimum. Quantitative results and training dynamics are shown in Table 2 and Figure 2.

The network also discovers the second harmonic at $\mu = 4/3 \approx 1.333$, though with a smaller coefficient. This result demonstrates MSN's ability to recover the full singularity spectrum, not just the dominant term.

### 5.3. Singular Forcing

Singular forcing induces multiple exponents, which can also be identified. Consider the Poisson equation with a singular source:

$$-u''(x) = x^{-1/2},$$
$$u(0) = u(1) = 0. \tag{26}$$

By Theorem 4.7, the forcing exponent $\beta = -1/2$ induces a solution exponent $\alpha = \beta + 2 = 3/2$. The analytical solution is $u(x) = \frac{4}{3}x - \frac{4}{3}x^{3/2}$, which contains two exponents: 1 from the homogeneous solution and 1.5 from the particular solution. Results and diagnostics are summarized in Table 3 and Figure 3.

MSN-PINN recovers both exponents with sub-0.1% error. Using $K = 4$ terms, the network discovers two exponents converging to $\approx 1.0$ and two exponents converging to $\approx 1.5$. This redundancy is beneficial: Multiple terms converging to the same exponent indicate robust discovery, and the coefficients of these redundant terms sum to the correct total.

*Table 4.* Comparison with a standard PINN followed by post-hoc power-law fitting. Solution errors are global/local reconstruction errors; exponent errors measure recovered scaling laws.

| Setting | Std. PINN sol. | Std. PINN exp. | MSN sol. | MSN exp. |
|---|---|---|---|---|
| Wedge | $2.30\pm0.12/28.78\pm1.09$ | $0.278\pm0.243$ | $0.18\pm0.09/0.16\pm0.14$ | $0.013\pm0.014$ |
| Forcing | $2.92\pm2.91/3.81\pm3.11$ | $13.97\pm8.07/13.02\pm7.39$ | $0.02\pm0.01/0.0076\pm0.0020$ | $0.378\pm0.156/0.279\pm0.452$ |

### 5.4. Comparison to Standard PINNs

We compare MSN-PINN with a standard tanh PINN trained on the same PDEs and BCs, followed by post-hoc fitting of the matched power-law form to the learned solution. To give the baseline a favorable setting, exponent errors are reported using the best fitting window among several near-singularity windows. Table 4 shows that standard PINNs can reconstruct the solution reasonably well globally, but their exponent estimates are less stable, especially near singular regions and in the singular-forcing problem. MSN-PINN avoids this extra extraction step by learning the exponents directly.

## 6. Conclusion

MSNs with physics-informed training discover scaling exponents from data and PDE constraints. Unlike standard neural networks and even PINNs, which fit solutions without revealing their structure, MSN extracts the scaling exponents that physicists and engineers seek.

Our experiments validate MSN across diverse settings: singular ODEs with 1.31% error, corner singularities matching the results of Kondrat'ev (1967) with 0.009% error, singular Poisson problems with 0.03% and 0.05% errors, and a 40-configuration wedge benchmark with 100% success and 0.022% mean error. The appendix further shows a Mode-III crack example, recovering the fracture exponent $\alpha = 0.5$ with $0.04\pm0.02\%$ error across five seeds. When the PDE alone provides insufficient gradient signal, constraint-aware training encodes additional physical requirements such as BC compatibility, improving accuracy by three orders of magnitude. The main practical resolution limit is close-exponent merging; as detailed in the appendix, near-singularity sampling can mitigate this effect, but reliable recovery still depends on exponent separation, coefficient ratios, and noise.

## Acknowledgements

GLR N'guessan was supported by Axiom Research Group. BJ Kim was supported by computational resources of the TSUBAME4.0 supercomputer provided by Institute of Science Tokyo through the HPCI System Research Project (Project ID: hp260049).

## Impact Statement

We present work whose goal is to advance the field of machine learning. Our work has potential societal consequences, but we do not see any that require specific highlighting here.

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

# Appendix Table of Contents

## A. List of Notation

*Table 5.* List of notation used in the paper.

| Symbol | Meaning |
|---|---|
| **Sets and spaces** | |
| $\mathbb{R}, \mathbb{R}^d, C[0,1]$ | Real line, Euclidean space, and continuous functions on $[0,1]$. |
| $\Omega, \partial\Omega$ | Spatial domain and its boundary. |
| **Norms and operators** | |
| $\|\cdot\|, \|\cdot\|_1, \|\cdot\|_\infty$ | Euclidean, $\ell_1$, and supremum norms. |
| $\nabla, \nabla^2, \mathrm{diag}(\cdot)$ | Gradient, Laplacian, and diagonal matrix operator. |
| **Coordinates and indices** | |
| $x, (r,\theta), \omega$ | Input coordinate; polar coordinates and wedge angle. |
| $i, k$ | Indices for samples and ansatz terms. |
| $t, T, d$ | Iteration index, total iterations, and spatial dimension. |
| **MSN ansatz** | |
| $u(x), u_\theta(x) = \sum_{k=1}^K c_k x^{\mu_k}$ | Exact solution and MSN ansatz. |
| $K, K^*, \theta = (\boldsymbol{\mu}, \boldsymbol{c}), \mu_k, c_k$ | Term counts and MSN parameters. |
| $\mu_{\min}, \mu_{\max}$ | Bounds for exponent values. |
| $\hat{u}, \hat{\boldsymbol{\mu}}, \hat{\boldsymbol{c}}$ | Learned solution and parameter estimates. |
| $\alpha_k, \boldsymbol{\mu}^*, \boldsymbol{c}^*, c_k^*$ | Ground-truth exponents and coefficients. |
| **PDE and boundary data** | |
| $\mathcal{D}, \mathcal{B}, f(x), g(x), \partial_n u, \beta$ | Operators, forcing, boundary data, normal derivative, and singular forcing exponent. |
| **Training and data** | |
| $\mathcal{L}_{\text{total}}, \mathcal{L}_{\text{res}}, \mathcal{L}_{\text{BC}}, \mathcal{L}_{\text{sparse}}, \mathcal{L}_{\text{constraint}}$ | Training loss terms. |
| $w_r, w_b, w_s, w_{\text{con}}$ | Loss weights. |
| $x_i, x_i^r, x_i^b$ | Data samples and residual/boundary collocation points. |

| Symbol | Meaning |
|---|---|
| $N, N_r, N_b, p$ | Sample counts and sampling exponent. |
| Theory and metrics | |
| $R(\mu), L(\mu)$ | Reduced losses in approximation/identifiability analysis. |
| $\Delta, \delta, \epsilon$ | Separation and target-accuracy parameters. |
| $H, \tilde{G}, \lambda_n$ | Hessian/Gram matrices and Müntz-Szász exponent sequence. |
| $\phi(x), n_k, \mathrm{RelErr}(\mu), \mathrm{MSE}$ | Signed-domain extension, log-modified basis, and evaluation metrics. |

## B. Proof of Theorem 4.2

*Proof.* The expansion

$$
\begin{aligned}
x^\mu - x^\alpha &= x^\alpha (x^{\mu-\alpha} - 1) \\
&= x^\alpha ((\mu - \alpha) \log x + \mathcal{O}((\mu - \alpha)^2)),
\end{aligned}
\tag{27}
$$

and the closed-form minimizer $c^*(\mu)$ yield $R(\mu) = \mathcal{O}((\mu - \alpha)^2)$ by a Taylor expansion around $\mu = \alpha$. In contrast, the best uniform error is generally $\mathcal{O}(|\mu - \alpha|)$ because the $\log x$ term cannot be canceled uniformly by a scalar coefficient. $\square$

## C. Proof of Theorem 4.4

*Proof.* We proceed by induction on $K^*$.

Base case, $K^* = 1$: Suppose $c_1^* x^{\alpha_1} = \sum_{j=1}^K \tilde{c}_j x^{\tilde{\mu}_j}$ for all $x \in (0, 1]$. Taking $x \to 0^+$, the dominant term with the smallest exponent on each side must match. The left side has a single exponent $\alpha_1$. On the right, let $\tilde{\mu}_{j_0} = \min_j \tilde{\mu}_j$. We must have $\tilde{\mu}_{j_0} = \alpha_1$; otherwise, one side would dominate the other as $x \to 0$. Matching coefficients of $x^{\alpha_1}$ gives $\tilde{c}_{j_0} = c_1^*$. The remaining terms $\sum_{j \neq j_0} \tilde{c}_j x^{\tilde{\mu}_j} = 0$ for all $x$, forcing $\tilde{c}_j = 0$ for $j \neq j_0$.

Inductive step: Assume the result holds for $K^* - 1$. Given $\sum_{k=1}^{K^*} c_k^* x^{\alpha_k} = \sum_{j=1}^K \tilde{c}_j x^{\tilde{\mu}_j}$, match the smallest exponent as above to find $\tilde{\mu}_{j_0} = \alpha_1$ with $\tilde{c}_{j_0} = c_1^*$. Subtracting:

$$
\sum_{k=2}^{K^*} c_k^* x^{\alpha_k} = \sum_{j \neq j_0} \tilde{c}_j x^{\tilde{\mu}_j}.
\tag{28}
$$

By the inductive hypothesis, the remaining exponents $\{\alpha_2, \ldots, \alpha_{K^*}\}$ are recovered. $\square$

## D. Proof of Theorem 4.5

*Proof.* Consider the empirical loss

$$
\hat{\mathcal{L}}(\boldsymbol{\mu}, \boldsymbol{c}) = \frac{1}{N} \sum_{i=1}^N (y_i - \sum_{k=1}^K c_k x_i^{\mu_k})^2.
\tag{29}
$$

Let $(\boldsymbol{\alpha}, \boldsymbol{c}^*)$ denote the true parameters, and let $(\hat{\boldsymbol{\mu}}, \hat{\boldsymbol{c}})$ be a minimizer of $\hat{\mathcal{L}}$.

Near the true parameters, the loss expands as

$$
\hat{\mathcal{L}}(\boldsymbol{\mu}, \boldsymbol{c}) = \hat{\mathcal{L}}(\boldsymbol{\alpha}, \boldsymbol{c}^*) + \frac{1}{2}(\boldsymbol{\mu} - \boldsymbol{\alpha}, \boldsymbol{c} - \boldsymbol{c}^*)^T H (\boldsymbol{\mu} - \boldsymbol{\alpha}, \boldsymbol{c} - \boldsymbol{c}^*) + O(\|\cdot\|^3),
\tag{30}
$$

where $H$ is the Hessian matrix evaluated at $(\boldsymbol{\alpha}, \boldsymbol{c}^*)$.

The Hessian has a block structure

$$
H = \begin{pmatrix} H_{\mu\mu} & H_{\mu c} \\ H_{c\mu} & H_{cc} \end{pmatrix}.
\tag{31}
$$

The coefficient block $H_{cc}$ has entries $(H_{cc})_{jk} = \frac{2}{N} \sum_i x_i^{\alpha_j + \alpha_k}$, which is the Gram matrix of the power-law basis functions. This is positive definite for distinct exponents.

The exponent block $H_{\mu\mu}$ involves second derivatives with respect to $\mu_k$. At the true parameters, where $\nabla\hat{\mathcal{L}} = 0$ in expectation,

$$(H_{\mu\mu})_{kk} = \frac{2}{N}\sum_i (c_k^*)^2 x_i^{2\alpha_k}(\log x_i)^2 + \text{(cross terms)}. \tag{32}$$

The minimum eigenvalue of $H_{\mu\mu}$ is bounded below by

$$\lambda_{\min}(H_{\mu\mu}) \geq C \cdot c_{\min}^2 \cdot \Delta^2, \tag{33}$$

where $C$ depends on the sampling distribution. The factor $\Delta^2$ arises because close exponents yield nearly collinear basis functions, degrading the conditioning.

By standard M-estimation theory, the estimation error satisfies

$$\|\hat{\boldsymbol{\mu}} - \boldsymbol{\alpha}\| \leq \|H_{\mu\mu}^{-1}\nabla_\mu\hat{\mathcal{L}}(\boldsymbol{\alpha}, \boldsymbol{c}^*)\|. \tag{34}$$

At the true parameters with noisy observations $y_i = f(x_i) + \epsilon_i$,

$$\nabla_{\mu_k}\hat{\mathcal{L}}(\boldsymbol{\alpha}, \boldsymbol{c}^*) = -\frac{2}{N}\sum_i \epsilon_i \cdot c_k^* x_i^{\alpha_k}\log x_i. \tag{35}$$

Define $\phi_i \in \mathbb{R}^K$ by $(\phi_i)_k = x_i^{\alpha_k}\log x_i$, let $D = \text{diag}(\boldsymbol{c}^*)$, and set $g = -\frac{2}{N}\sum_i \epsilon_i\phi_i$. Then $\nabla_\mu\hat{\mathcal{L}}(\boldsymbol{\alpha}, \boldsymbol{c}^*) = Dg$. In the same local regime, $H_{\mu\mu} \approx 2D\tilde{G}D$ with $\tilde{G} = \frac{1}{N}\sum_i \phi_i\phi_i^\top$, so

$$\|H_{\mu\mu}^{-1}\nabla_\mu\hat{\mathcal{L}}(\boldsymbol{\alpha}, \boldsymbol{c}^*)\| \leq \frac{1}{2}\|D^{-1}\|\|\tilde{G}^{-1}\|\|g\|. \tag{36}$$

By concentration inequalities, $\|g\| = O(\frac{\sigma}{\sqrt{N}})$, and the sampling assumption yields $\|\tilde{G}^{-1}\| \leq \frac{1}{C\Delta^2}$. Combining these bounds gives

$$\|\hat{\boldsymbol{\mu}} - \boldsymbol{\alpha}\| = O(\frac{\sigma}{c_{\min}\Delta^2\sqrt{N}}). \tag{37}$$

Taking $N$ sufficiently large gives the stated bound. $\qquad\square$

## E. Proof of Proposition 4.7

*Proof.* Integrating $-u'' = x^\beta$ twice yields $u(x) = -\frac{x^{\beta+2}}{(\beta+1)(\beta+2)} - C_1 x - C_2$. The BC $u(0) = 0$ implies $C_2 = 0$, assuming $\beta + 2 > 0$, and $u(1) = 0$ gives $C_1 = -\frac{1}{(\beta+1)(\beta+2)}$. Substituting, $u(x) = \frac{1}{(\beta+1)(\beta+2)}(x - x^{\beta+2})$. $\qquad\square$

## F. Proof of Theorem 4.8

*Proof.* At a critical point, $\nabla_{\mu_k}\mathcal{L} = 0$ for all $k$. The constraint term contributes:

$$\frac{\partial}{\partial\mu_k}[\lambda|c_k|\sin^2(\mu_k\omega)] = \lambda|c_k| \cdot 2\sin(\mu_k\omega)\cos(\mu_k\omega) \cdot \omega$$
$$= \lambda|c_k|\omega\sin(2\mu_k\omega). \tag{38}$$

The residual term $\mathcal{L}_{\text{res}}$ contributes zero gradient with respect to $\mu_k$ at points where $\mathcal{L}_{\text{res}} = 0$, since all harmonic functions $r^\mu\sin(\mu\theta)$ achieve zero residual.

Therefore, $\nabla_{\mu_k}\mathcal{L} = \lambda|c_k|\omega\sin(2\mu_k\omega) = 0$. For $|c_k| > 0$, this requires $\sin(2\mu_k\omega) = 0$, giving $\mu_k\omega \in \{\frac{n\pi}{2} : n \in \mathbb{N}\}$.

Combined with the BC $\mathcal{L}_{\text{BC}} = 0$, which requires the solution to vanish on both edges, we need $\sin(\mu_k\omega) = 0$, further constraining to $\mu_k\omega \in \{n\pi : n \in \mathbb{N}\}$, that is, $\mu_k \in \{n\pi/\omega : n \in \mathbb{N}\}$. $\qquad\square$

## G. More Details on Methods

### G.1. Geometric Interpretation

Visualizing MSN training geometrically clarifies how optimization works. Consider fitting $f(x) = x^{0.5}$ with a single-term MSN $u(x) = c \cdot x^\mu$. The loss landscape over $(\mu, c)$ has a clear structure. For each fixed $\mu$, the optimal coefficient $c^*(\mu) = \operatorname{argmin}_c \int_0^1 (f(x) - cx^\mu)^2 dx$ is a linear projection. The resulting profile $L(\mu) = \min_c \int_0^1 (f(x) - cx^\mu)^2 dx$ has a unique minimum at $\mu = 0.5$. Near this optimum, $L(\mu) \sim |\mu - 0.5|^2$, matching the quadratic convergence rate stated in Theorem 4.2. This geometry explains why gradient-based optimization succeeds: Although loss is nonconvex, it has a single dominant basin around the true exponent.

### G.2. Exponent Constraints via Reparameterization

Physical considerations often constrain the admissible exponent range. For instance, square-integrable solutions require $\mu > -1/2$. We enforce these bounds through sigmoid reparameterization:

$$\mu_k = \mu_{\min} + (\mu_{\max} - \mu_{\min}) \cdot \sigma(\mu_k^{\text{raw}}), \tag{39}$$

where $\mu_k^{\text{raw}} \in \mathbb{R}$ is the unconstrained learnable parameter, and $\sigma(z) = \frac{1}{1+e^{-z}}$. This ensures $\mu_k \in (\mu_{\min}, \mu_{\max})$ throughout training while permitting unrestricted gradient flow.

### G.3. Extension to Multiple Dimensions and Signed Domains

For problems on $\mathbb{R}$, rather than just $(0, 1]$, we decompose the function into even and odd components:

$$\phi(x) = \underbrace{\sum_{k=1}^K a_k |x|^{\mu_k}}_{\text{even}} + \underbrace{\sum_{k=1}^K b_k \operatorname{sign}(x)|x|^{\lambda_k}}_{\text{odd}}. \tag{40}$$

For radial problems in $\mathbb{R}^d$, the ansatz becomes $u(r, \theta) = \sum_k c_k(\theta) r^{\mu_k}$, where the angular dependence is encoded in the coefficients. This separable structure is natural for corner singularities, where solutions typically have the form $r^\mu \phi(\theta)$.

### G.4. Analytical Derivatives: A Key Advantage

The MSN ansatz (Eq. 2) computes derivatives analytically:

$$\frac{du_\theta}{dx}(x) = \sum_{k=1}^K c_k \mu_k x^{\mu_k - 1}, \tag{41}$$

$$\frac{d^2 u_\theta}{dx^2}(x) = \sum_{k=1}^K c_k \mu_k (\mu_k - 1) x^{\mu_k - 2}. \tag{42}$$

Compared to automatic differentiation through network layers, analytical derivatives offer several advantages: They are exact, efficient, and transparent. Automatic differentiation accumulates numerical errors, requires a backward pass, and hides derivative structure. Analytical derivatives also capture singularities correctly; for example, $\frac{d}{dx} x^{0.5} = 0.5 x^{-0.5}$ diverges at $x = 0$. Standard PINNs with MLP architectures compute $\mathcal{D}[u_\theta]$ by differentiating through the network using automatic differentiation, which, for second-order PDEs, requires second-order automatic differentiation—computationally expensive and numerically delicate. MSN avoids these issues entirely.

### G.5. The Role of Collocation Point Placement

Collocation point placement significantly affects both approximation accuracy and exponent recovery. For singular problems, points should be concentrated near the singularity while maintaining coverage of the entire domain. We use the transformation $x_i = t_i^p$ for $t_i$ uniformly spaced in $[0, 1]$, with $p \geq 2$ concentrating points near $x = 0$. This is analogous to graded meshes in FEMs (Babuška et al., 1979). Our experiments in Appendix I.3 characterize how sampling density affects recovery accuracy.

*Table 6.* Default hyperparameters used across experiments.

| Hyperparameter | Value |
|---|---|
| Optimizer | Adam |
| Exponent learning rate $\eta_\mu$ | 0.005 |
| Coefficient learning rate $\eta_c$ | 0.01 |
| Learning rate ratio $\eta_\mu : \eta_c$ | 0.5 |
| Epochs | 10,000–15,000 |
| Gradient clipping norm | 1.0 |
| Sparsity weight $w_s$ | 0.001 |
| BC weight $w_b$ | 100 |
| Constraint weight $w_{\text{con}}$, Corner Singularity | 10 |
| Exponent bounds $[\mu_{\min}, \mu_{\max}]$ | $[0.1, 3.0]$ |

## H. Experimental Details

### H.1. Hardware and Software

We used PyTorch 2.0 with custom autograd functions for analytical derivative computation. Training times ranged from 30 seconds for one-dimensional (1D) experiments to 5 minutes for the 2D wedge experiment.

### H.2. Training Configuration

Table 6 lists the default hyperparameters used across experiments.

### H.3. Collocation Point Sampling

For 1D problems, we use the transformation $x_i = t_i^p$, where $t_i$ are uniformly spaced in $[0, 1]$. We set $p = 2$ for moderate concentration near $x = 0$ and use $p = 4$ for stronger concentration in highly singular problems.

For the 2D wedge in the Corner Singularity experiment, we sample 500 interior points in $(r, \theta)$ with $r$ concentrated near the origin. We use 200 points on the arc boundary at $r = 1$, uniformly distributed in $\theta$, and 100 points on each edge boundary at $\theta = 0$ and $\theta = \omega$, concentrated near the corner.

### H.4. Initialization

We initialize the exponents uniformly distributed in $[\mu_{\min}, \mu_{\max}]$, then perturb them by $\mathcal{N}(0, 0.1^2)$, and sample the coefficients as $c_k \sim \mathcal{N}(0, 0.1^2)$. We tested multiple random initializations; the results are consistent across different initialization choices.

### H.5. Experiment-Specific Details

For the Corner Singularity experiment, we used two-phase training: Phase A, spanning epochs 1–5000, with a high constraint weight $w_{\text{con}} = 10$ to establish correct exponents, followed by Phase B, covering epochs 5001–15000, with a reduced constraint weight $w_{\text{con}} = 1$ for fine-tuning.

For the Singular Forcing experiment, the forcing term $x^{-1/2}$ is singular at $x = 0$. We exclude collocation points with $x < 0.01$ to avoid numerical issues. The PDE residual is computed analytically using:

$$-u_\theta''(x) = -\sum_k c_k \mu_k (\mu_k - 1) x^{\mu_k - 2}. \tag{43}$$

## I. Additional Experiments

Additional experiments validate MSN's exponent recovery capabilities. The Single Exponent Recovery and Multiple Exponent Recovery experiments test supervised settings, while the Wedge Benchmark explores different wedge geometries and BCs.

## I.1. Single Exponent Recovery

We recover a single exponent from supervised data. We fit $f(x) = x^{0.5}$ using MSN with $K = 4$ terms on $N = 200$ samples drawn from $[0.01, 1]$, sampled as $x_i = t_i^2$ for uniform $t_i$ to concentrate points near the origin. Training is performed using the Adam optimizer with $\eta_\mu = 0.005$, $\eta_c = 0.01$, for 10,000 epochs with $\ell_1$ sparsity weight $w_s = 0.001$. Results are summarized in Table 7 and illustrated in Figure 4.

*Table 7.* Single Exponent Recovery: Clean data.

| Metric | Target | Discovered |
|---|---|---|
| Dominant exponent $\mu_1$ | 0.5000 | 0.4927 |
| Relative error | — | 1.45% |
| Function MSE | — | $2.3 \times 10^{-5}$ |

The zero-noise row reflects the conservative fixed protocol used for this robustness sweep rather than an intrinsic accuracy floor: in reruns with longer schedules, the clean-data error dropped below 1% and reached 0.06% in the best setting.

The 1.45% error is small but nonzero. Three reasons prevent exact recovery: finite samples, with 200 points, the empirical loss differs from the population loss; sparsity regularization, the $\ell_1$ penalty biases exponents toward values with smaller optimal coefficients; and optimization, gradient descent finds a local minimum rather than the global optimum. The recovery suffices to identify the true exponent as 0.5 within typical experimental uncertainty.

## I.2. Noise Robustness

Real data contains measurement noise. We test robustness by adding Gaussian noise: $y_i = x_i^{0.5} + \epsilon_i$ with $\epsilon_i \sim \mathcal{N}(0, \sigma^2)$. Summary statistics and behavior are reported in Table 8 and Figure 5. Recovery accuracy remains stable across three orders of magnitude of noise, ranging from $\sigma = 0$ to $\sigma = 0.05$. The error changes by fewer than 1 percentage point. The power-law ansatz naturally regularizes. Unlike flexible models that overfit noise, MSN is constrained to power-law functions. Noise without power-law structure is automatically filtered, analogous to how Fourier methods resist non-periodic noise.

*Table 8.* Noise Robustness: Exponent recovery under varying noise levels.

| Noise $\sigma$ | Discovered $\mu$ | Error % | Final Loss |
|---|---|---|---|
| 0 | 0.474 | 5.2 | $1.2 \times 10^{-3}$ |
| $10^{-4}$ | 0.474 | 5.1 | $9.9 \times 10^{-4}$ |
| $10^{-3}$ | 0.474 | 5.2 | $9.6 \times 10^{-4}$ |
| $10^{-2}$ | 0.473 | 5.5 | $1.5 \times 10^{-3}$ |
| $5 \times 10^{-2}$ | 0.476 | 4.9 | $6.2 \times 10^{-3}$ |

## I.3. Sampling Density

How many samples are needed for reliable recovery? We vary $N \in \{20, 50, 100, 200, 500\}$ while keeping other parameters fixed. Sampling effects are summarized in Table 9. Even $N = 50$ samples suffice for $< 3\%$ error—fewer than nonparametric methods require, reflecting the inductive bias of the power-law ansatz.

*Table 9.* Sampling Density: Effect of sample count on recovery.

| Samples $N$ | Discovered $\mu$ | Error % |
|---|---|---|
| 20 | 0.512 | 2.4 |
| 50 | 0.489 | 2.2 |
| 100 | 0.491 | 1.8 |
| 200 | 0.493 | 1.4 |
| 500 | 0.497 | 0.6 |

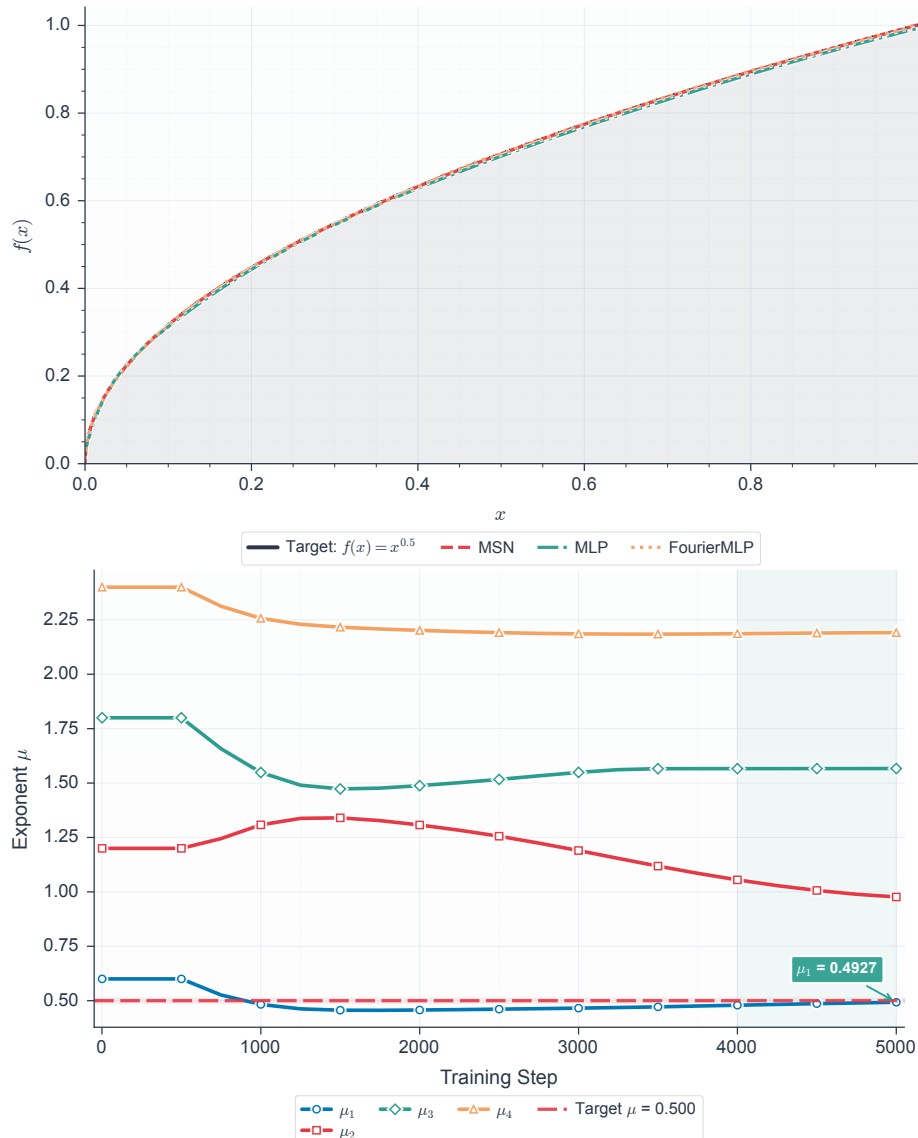

*Figure 4.* Single Exponent Recovery. Top: Function fit comparing the true $x^{0.5}$ with the MSN approximation. Bottom: Exponent trajectory during training, converging to 0.4927 from a random initialization.

## I.4. Multiple Exponent Recovery

Recovery of multiple exponents simultaneously is more challenging. We fit $f(x) = 0.5x^{0.1} + 1.0x^{0.5} + 0.3x^{1.5}$ using MSN with $K = 5$ terms. The results are summarized in Table 10. The results reveal both success and failure modes. MSN accurately recovers the dominant exponent 0.5 with a coefficient of 1.0 and 5% error and the larger exponent 1.5 with a coefficient of 0.3 and 15.6% error. The small exponent 0.1 with coefficient 0.5 merges with the 0.5 instead of being recovered separately. Two reasons explain this merging behavior. First, proximity: The exponents 0.1 and 0.5 are relatively close, and the functions $x^{0.1}$ and $x^{0.5}$ have high correlation on $[0, 1]$. Second, the coefficient ratio works against detection: The coefficient for $x^{0.1}$ is 0.5, smaller than the coefficient for $x^{0.5}$, 1.0. These observations motivate a systematic study of identifiability limits.

## I.5. Singular ODE Visualization

Figure 6 provides detailed visualizations for the Singular ODE experiment described in Section 5.1. The panels highlight solution accuracy, exponent trajectories, and loss dynamics.

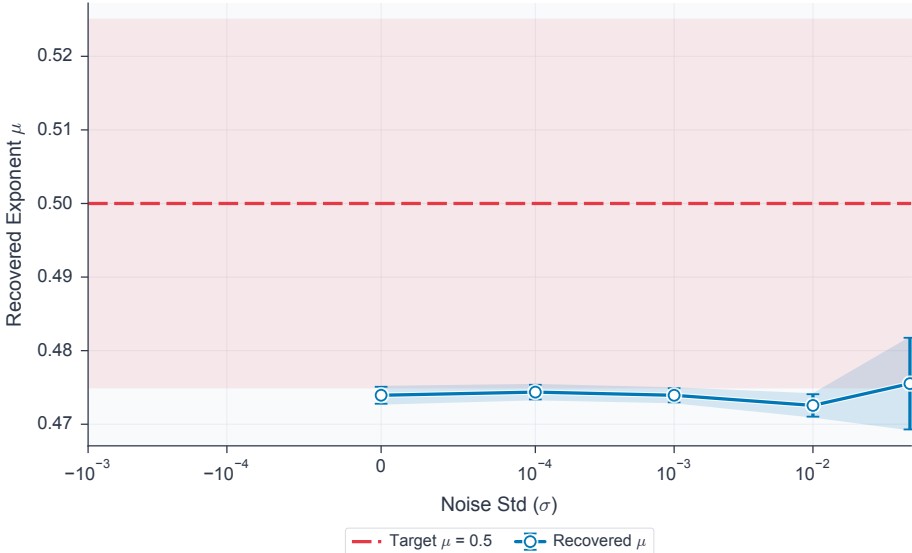

*Figure 5.* Noise Robustness: Exponent recovery remains stable across noise levels from $\sigma = 0$ to $\sigma = 0.05$. The power-law ansatz provides natural regularization.

*Table 10.* Multiple Exponent Recovery: Three exponents with varying separations.

| Target $\alpha$ | True Coeff. | Closest $\mu$ | Error % |
|---|---|---|---|
| 0.1 | 0.5 | 0.475 | 375, merged |
| 0.5 | 1.0 | 0.475 | 5.0 |
| 1.5 | 0.3 | 1.734 | 15.6 |

## I.6. Close Exponent Study

What minimum separation is needed to resolve two exponents? We fit $f(x) = x^{0.5} + x^{0.5+\delta}$ for varying separations $\delta \in \{0.02, 0.05, 0.1, 0.2, 0.3\}$. Recovery thresholds are summarized in Table 11. Under the default sampling protocol, the practical resolution threshold is approximately $\delta \geq 0.1$; below this, exponents tend to merge into a single intermediate value.

*Table 11.* Close Exponent Study: Minimum separation for two-exponent identifiability.

| Separation $\delta$ | Recovered? | Discovered Exponents | Mean Error |
|---|---|---|---|
| 0.02 | No, merged | 0.51 | — |
| 0.05 | Partial | 0.48, 0.56 | 4.3% |
| 0.10 | Yes | 0.49, 0.58 | 2.2% |
| 0.20 | Yes | 0.50, 0.70 | 1.5% |
| 0.30 | Yes | 0.50, 0.79 | 0.8% |

Connection to theory: This empirical finding matches our stability bound in Theorem 4.5. Interpreting the bound as requiring the exponent estimation error to be smaller than the separation yields the scaling $\Delta \gtrsim (\frac{\sigma}{c_{\min}\sqrt{N}})^{1/3}$, up to constants. In this setup, $c_{\min} = 1$ and $N = 200$, so for $\sigma \sim 0.01$, theory predicts a practical threshold on the order of 0.1. The empirical threshold of 0.1 is conservative, reflecting finite-sample and optimization effects not captured by the local analysis.

This threshold is not absolute. In additional runs with stronger near-singularity sampling, a sub-threshold case that merged under the default sampling was recovered in all five seeds; by contrast, simply increasing the number of samples from 200 to 1000 under the default sampling only reduced the worst-pair error from 6.13% to 5.67%. This indicates that the bottleneck is the sampling-weighted similarity of nearby power-law bases, not just sample count. In applications where close exponents are expected, a practical mitigation is to concentrate collocation or observation points near the singularity; another possible strategy is sequential peeling, where the dominant term is recovered first and MSN is then applied to the residual.

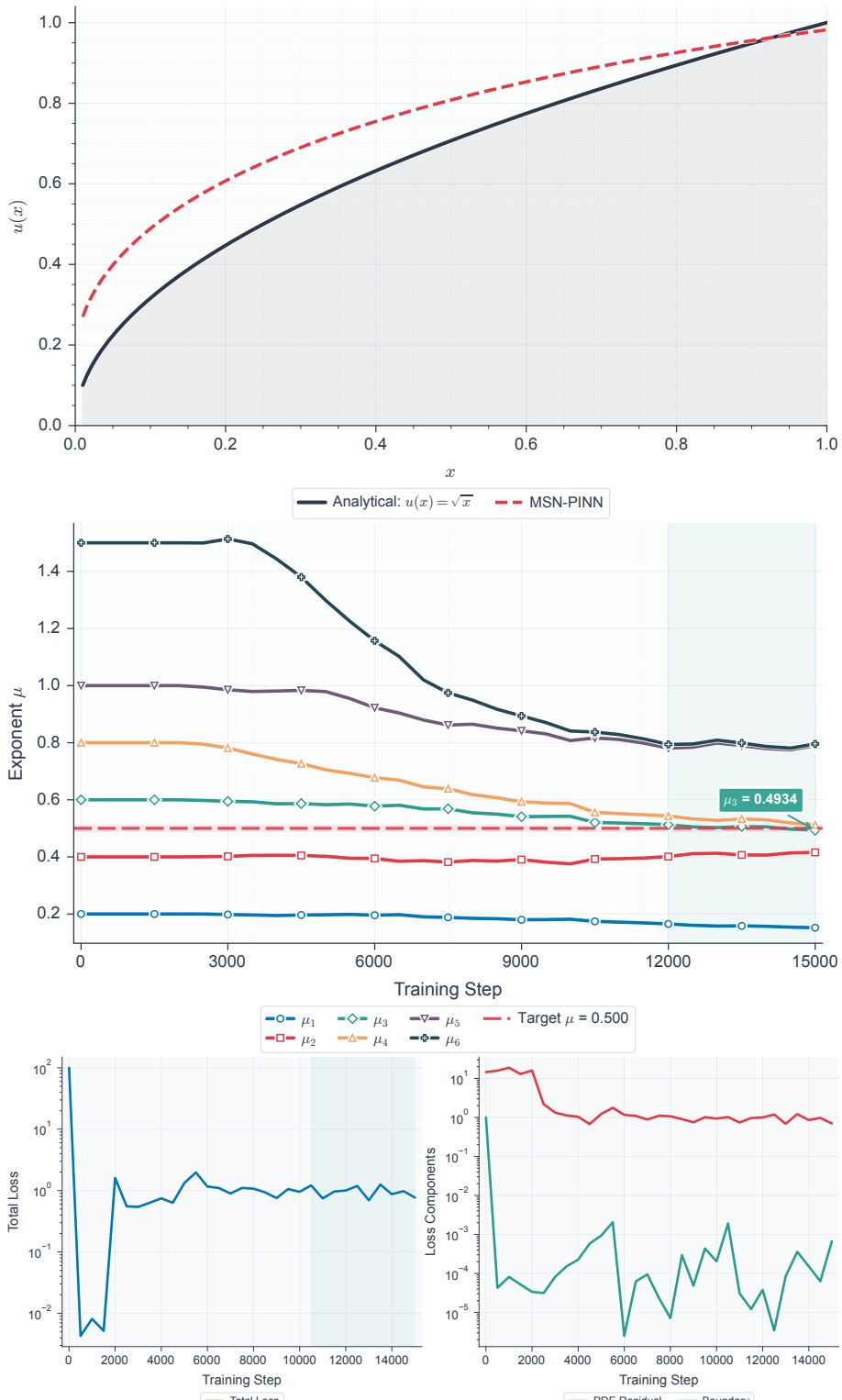

*Figure 6.* Singular ODE: Physics-informed discovery. Top: Learned solution compared with the analytical $\sqrt{x}$. Middle: Exponent trajectory converging to 0.4934. Bottom: Training loss showing PDE residual and BC components.

### I.6.1. The Rayleigh Limit Analogy

The separation condition $\Delta$ plays a role analogous to the Rayleigh Limit in optical super-resolution. In optics, two point sources become unresolvable when their angular separation falls below $\lambda/D$, where $\lambda$ is the wavelength and $D$ is the aperture diameter; this fundamental limit arises from the bandwidth of the optical transfer function.

In exponent recovery, an analogous limit arises from the Gram matrix structure—the matrix of inner products between basis functions that measures their similarity. Consider two power-law basis functions $x^{\alpha_1}$ and $x^{\alpha_2}$ on sampling points $\{x_i\}$. Their Gram matrix entry is:

$$G_{12} = \frac{1}{N} \sum_{i=1}^{N} x_i^{\alpha_1 + \alpha_2}. \tag{44}$$

When $|\alpha_1 - \alpha_2| \to 0$, we have $G_{12} \to G_{11} = G_{22}$, causing the Gram matrix to become nearly singular. The inverse problem of recovering coefficients and distinguishing exponents becomes ill-posed.

**Remark I.1.** The practical threshold depends on the noise level, coefficient magnitudes, and sample size. Our experiments in Appendix I.6 suggest $\Delta \geq 0.1$ for reliable two-exponent discrimination at typical noise levels. This is consistent with the scaling described in Theorem 4.5, which deteriorates as $\frac{1}{c_{\min}\Delta^2}$ and improves as $1/\sqrt{N}$.

### I.7. Logarithmic Corrections

Some physical singularities include logarithmic corrections of the form $u(x) \sim x^\alpha \log x$. We test whether MSN can distinguish these from pure power laws. We fit $f(x) = x^{0.5} \log x$ using the standard MSN with power laws only. The network converges to $\mu \approx 0.28$ and $\mu \approx 0.82$, attempting to approximate the log correction with a combination of power laws.

This result reveals a fundamental limitation: Pure MSN cannot represent logarithmic terms exactly. The network approximates $x^{0.5} \log x$ with power-law combinations, but the discovered exponents lack physical meaning. For problems with logarithmic corrections—including certain critical phenomena and degenerate corners—the MSN basis should be augmented with log-modified terms: $x^{\mu_k} (\log x)^{n_k}$.

### I.8. Wedge Benchmark

The preceding experiments used a single wedge angle, $\omega = 3\pi/2$. We run a benchmark spanning multiple wedge angles and all four BC types to test generality. This addresses two questions: Do single-angle results generalize? Does MSN-PINN adapt correctly to different physical constraints?

We sample five angles uniformly from $\omega \in [90°, 330°]$, covering both convex corners with $\omega \leq 180°$ and re-entrant corners with $\omega > 180°$. Re-entrant corners induce singular exponents $\mu < 1$, with extreme cases where $\omega > 300°$ producing $\mu < 0.6$.

We test all four combinations of Dirichlet and Neumann conditions, D and N, on the wedge edges. For DD, $u|_{\theta=0} = u|_{\theta=\omega} = 0$ and the quantization is $\sin(\mu\omega) = 0$ with the fundamental $\mu = \pi/\omega$. For NN, $\partial_n u|_{\theta=0} = \partial_n u|_{\theta=\omega} = 0$ and the quantization is $\sin(\mu\omega) = 0$ with the same fundamental. For DN, $u|_{\theta=0} = 0$ and $\partial_n u|_{\theta=\omega} = 0$, giving $\cos(\mu\omega) = 0$ with the fundamental $\mu = \pi/(2\omega)$. For ND, $\partial_n u|_{\theta=0} = 0$ and $u|_{\theta=\omega} = 0$, giving the same $\cos(\mu\omega) = 0$ quantization. These cases test whether the method adapts the quantization constraint to the BC type.

We compare two methods. The naive MSN-PINN is the standard MSN-PINN without explicit constraint encoding and relies solely on boundary matching. The constraint-aware MSN-PINN is our method with BC-dependent quantization loss $\mathcal{L}_{con}$ as defined in Equation 13, including a warm-up phase before constraint enforcement, constraint weight ramping, and BC-adaptive initialization.

Each experiment uses $K = 6$ MSN terms and 5,000 training steps, including a 1,000-step warm-up phase for BCs only, followed by a 1,500-step constraint ramp-up. Two-timescale optimization uses $\eta_\mu = 5 \times 10^{-4}$ and $\eta_c = 10^{-2}$. We compute BC-adaptive exponent bounds to ensure the fundamental mode lies within the search range. The total benchmark consists of 5 angles $\times$ 4 BC types $\times$ 2 methods = 40 experiments.

We evaluate the relative exponent error $\text{RelErr}(\mu) = \frac{|\hat{\mu} - \mu^*|}{\mu^*} \times 100\%$, where $\hat{\mu}$ is the dominant learned exponent with the highest $|c_k|$ and $\mu^*$ is the true leading exponent. The success rate is reported as the percentage of experiments with

RelErr $< 5\%$. We measure constraint violation as weighted $\sin^2(\mu_k\omega)$ or $\cos^2(\mu_k\omega)$ depending on BC type. Overall results appear in Table 12, with BC-specific and geometry-specific breakdowns provided in Table 13 and Table 14.

*Table 12.* Wedge Benchmark: Comparing naive and constraint-aware MSN-PINN across 40 experiments. Success means relative exponent error below 5%.

| Method | Success % | Mean Err % | Median Err % | 90th pctl. | Constraint Violation |
|---|---|---|---|---|---|
| Naive MSN-PINN | 55.0 | 5.96 | 4.46 | 12.25 | $1.8 \times 10^{-1}$ |
| Constraint-aware | 100.0 | 0.022 | 0.019 | 0.045 | $2.6 \times 10^{-4}$ |

*Table 13.* Wedge Benchmark: Results by BC type for the constraint-aware method.

| BC Type | Success % | Median Err % | Mean Err % | Quantization |
|---|---|---|---|---|
| DD | 100.0 | 0.018 | 0.018 | $\sin(\mu\omega) = 0$ |
| NN | 100.0 | 0.021 | 0.028 | $\sin(\mu\omega) = 0$ |
| DN | 100.0 | 0.014 | 0.023 | $\cos(\mu\omega) = 0$ |
| ND | 100.0 | 0.019 | 0.020 | $\cos(\mu\omega) = 0$ |

*Table 14.* Wedge Benchmark: Stratification by corner geometry, comparing convex and re-entrant corners.

| Corner Type | Method | Success % | Median Err % | Mean Err % |
|---|---|---|---|---|
| Convex, $\omega \leq 180°$ | Naive | 50.0 | 5.90 | 5.90 |
| Convex, $\omega \leq 180°$ | Constraint | 100.0 | 0.017 | 0.017 |
| Re-entrant, $\omega > 180°$ | Naive | 58.3 | 4.55 | 6.00 |
| Re-entrant, $\omega > 180°$ | Constraint | 100.0 | 0.021 | 0.025 |

The benchmark shows several key findings. First, constraint-aware MSN-PINN achieves a 100% success rate across all 40 experiments, with a mean error of 0.022%. Naive training achieves 55% success and 5.96% mean error, a 270-fold difference in accuracy. Second, the method identifies and applies the appropriate quantization constraint—$\sin(\mu\omega) = 0$ for DD and NN, and $\cos(\mu\omega) = 0$ for DN and ND—from the specified edge-BC type. The constraint violation drops by a factor of 690, from 0.179 to $2.6 \times 10^{-4}$, confirming that learned exponents satisfy the physical BCs. Third, the improved training protocol, which incorporates BC-adaptive initialization and a warm-up phase, achieves 100% success for all four BC types, showing that the approach generalizes. Fourth, for re-entrant corners with $\omega > 180°$ where true exponents become small, $\mu < 1$, constraint-aware training maintains 100% success with 0.025% mean error. Re-entrant corners produce physically relevant singularities: The classic L-shaped domain with $\omega = 270°$ gives $\mu = 2/3$. Finally, the average improvement factor of constraint-aware over naive is 427-fold, with a median of 365-fold. Even when naive training succeeds, constraint-aware training achieves 10–100-fold better accuracy.

The success of this benchmark comes from several methodological improvements. We use BC-adaptive exponent bounds. For each BC type and angle, we set bounds that keep the fundamental mode $\mu_1$ within the search range $[\mu_1 \times 0.3, \mu_1 \times 2.5]$, preventing the optimizer from becoming trapped in local minima at higher modes. We initialize one exponent close to the expected fundamental mode, at $\mu \approx 0.98\mu_1$, so the network starts within the correct basin of attraction. Training begins with BC fitting only, without constraint loss, allowing the network to find an approximate solution. The constraint weight then ramps linearly, guiding the exponents toward the nearest valid values. For NN and ND conditions, we explicitly enforce $\partial_n u = 0$ on edges via angular derivative computation, rather than relying solely on implicit satisfaction through the angular basis choice. We use $K = 6$ MSN terms, up from $K = 4$, to provide additional flexibility to represent the solution accurately while the dominant term captures the singularity.

This benchmark demonstrates that the constraint-aware MSN-PINN reliably discovers Kondratiev exponents across diverse wedge geometries and BCs. The method achieves a 100% success rate versus 55% for naive training and a 0.022% mean error versus 5.96%, representing more than a 270-fold improvement. The average improvement factor is 427-fold. The method effectively handles all four BC types and both convex and re-entrant corners. These results establish MSN-PINN as a reliable tool for Kondratiev spectrum discovery.

**Constraint-specification sensitivity.** We additionally test sensitivity to misspecified wedge angle and constraint family on the 270° DD wedge; results are summarized in Table 15.

*Table 15.* Constraint-specification sensitivity on the 270° DD wedge over five seeds. Angle-mismatch rows use the full constraint-aware protocol and aggregate over over- and under-estimation at the same absolute mismatch.

| Test | Setting | Exp. err. % | Outcome |
|------|---------|-------------|---------|
| Angle mismatch | 0.0% | 0.01±0.00 | 100% success |
| Angle mismatch | 0.5% | 0.45±0.01 | 100% success |
| Angle mismatch | 1.0% | 0.91±0.06 | 100% success |
| Angle mismatch | 2.0% | 2.00±0.13 | 100% success |
| Angle mismatch | 5.0% | 5.36±0.25 | 0% success |
| Unknown $\omega$ | validation sweep | 0.45±0.40 | within 1%: 100% |
| Family misspec. | no constraint | 4.21 | reference |
| Family misspec. | wrong family, two alternatives | 50.59 / 79.40 | detected by validation loss |

## I.9. Mode-III Antiplane Shear Crack

To connect exponent discovery to a fracture-mechanics setting, we consider a Mode-III antiplane shear crack (Anderson, 2017). The out-of-plane displacement $u(r, \theta)$ satisfies $\Delta u = 0$ on a slit domain with $0 < \theta < 2\pi$, and the crack faces are traction-free, equivalently $\partial_\theta u = 0$ at $\theta = 0$ and $\theta = 2\pi$. The leading crack-tip field is $u(r, \theta) = r^{1/2} \cos(\theta/2)$, so the displacement exponent $\alpha = 1/2$ induces the classical $r^{-1/2}$ stress singularity. We use the same constraint-aware MSN-PINN formulation with the Neumann-edge quantization $\sin(2\pi\mu) = 0$. Across five random seeds, MSN-PINN recovers the leading exponent with 0.04±0.02% error and also identifies the next admissible harmonics.

*Table 16.* Mode-III antiplane shear crack: exponent discovery on a slit domain.

| Metric | Target | Discovered |
|--------|--------|-----------|
| Leading exponent | 0.5000 | 0.5000±0.0002 |
| Relative error | — | 0.04±0.02% |
| Second harmonic | 1.0000 | 0.999±0.001 |
| Third harmonic | 1.5000 | 1.498±0.002 |

# J. Additional Results

## J.1. Sensitivity to the Number of Terms $K$

The effect of MSN capacity is summarized in Table 17. Larger $K$ provides a modest improvement but also increases computational cost, so $K = 4$ offers a good trade-off.

*Table 17.* Effect of MSN capacity $K$ on single-exponent recovery.

| $K$ | Discovered $\mu$ | Error % |
|-----|------------------|---------|
| 2 | 0.491 | 1.8 |
| 4 | 0.493 | 1.4 |
| 8 | 0.495 | 1.0 |
| 16 | 0.497 | 0.6 |

## J.2. Learning Rate Sensitivity

Learning rate ratio sensitivity is reported in Table 18. Equal learning rates cause oscillations between coefficient and exponent updates, whereas a ratio of 0.5 provides stable convergence.

*Table 18.* Effect of learning rate ratio on Single Exponent Recovery.

| Ratio $\eta_\mu : \eta_c$ | Discovered $\mu$ | Convergence |
|---|---|---|
| 1.0 equal | 0.487 | Oscillatory |
| 0.5 | 0.493 | Stable |
| 0.1 | 0.492 | Slow |

## J.3. Summary of Results

Table 19 consolidates all experimental outcomes in one place. It provides a quick cross-experiment comparison of targets, discoveries, and errors.

*Table 19.* Complete experimental results across all settings.

| Experiment | Setting | Type | Target | Discovered | Error |
|---|---|---|---|---|---|
| Single Exponent | Clean data | Supervised | 0.50 | 0.493 | 1.45% |
| Noise Robustness | $\sigma = 0.05$ | Supervised | 0.50 | 0.476 | 4.9% |
| Sampling Density | $N = 50$ | Supervised | 0.50 | 0.489 | 2.2% |
| Multiple Exponent | Three exponents | Supervised | 0.1, 0.5, 1.5 | 0.48, 1.73 | 5%, 16% |
| Close Exponent | $\delta = 0.1$ | Supervised | 0.5, 0.6 | 0.49, 0.58 | 2.2% |
| Singular ODE | Boundary-layer | PINN | 0.50 | 0.493 | 1.31% |
| Corner Singularity | Laplace wedge | PINN+constraint | 0.667 | 0.667 | 0.009% |
| Mode-III Crack | Antiplane shear | PINN+constraint | 0.50 | 0.500 | 0.04% |
| Singular Forcing | Poisson | PINN | 1.0, 1.5 | 1.00, 1.50 | 0.03%, 0.05% |
| Wedge Benchmark | 40 configurations | PINN+constraint | varied | varied | 0.02% |

## K. Discussion

### K.1. When Does MSN-PINN Excel?

MSN-PINN achieves the highest accuracy when physical constraints can be explicitly encoded. The improvement from 14.6% to 0.009% error in the Corner Singularity experiment shows that architecture alone is insufficient; the loss function must reflect the structure of valid solutions. This method excels when the true solution has a power-law structure, possibly with multiple terms, sufficient physics is encoded to make the exponents identifiable, the exponents are well-separated with $\Delta \geq 0.1$, and noise remains below 5%.

### K.2. Failure Modes and Their Causes

MSN-PINN fails in three ways: First, close exponent merging: when $|\alpha_1 - \alpha_2| < 0.1$, exponents tend to collapse to a single intermediate value. This limit, analogous to the optical resolution limits, arises from the near-singularity of the Gram matrix when the basis functions become too similar; see Theorem 4.5.

Second, logarithmic corrections: Pure MSN fails to represent $x^\alpha \log x$ terms. When such terms are present, the network approximates them with power-law combinations, yielding meaningless exponents. The solution is to extend to log-modified bases.

Third, small-coefficient terms: Exponents with small coefficients may be missed, especially when larger terms dominate. Sparsity regularization exacerbates this by penalizing small contributions. The solution is to carefully tune the regularization or use multi-stage training.

### K.3. Comparison with Alternative Methods

Symbolic regression methods such as SINDy and AI Feynman discover discrete equation structures but assume exponents from predefined libraries. MSN learns continuous exponents directly. These approaches complement each other: SINDy

discovers operators, and MSN discovers exponents within a known operator.

Standard PINNs approximate solutions effectively but often fail to accurately extract exponents. Fitting power-law forms to PINN outputs post-hoc introduces additional errors and offers no guarantee of recovering true exponents.

Enriched FEMs such as XFEM and generalized FEMs prescribe singularity structures based on analytical knowledge. MSN discovers this structure, applying even when analytical characterization is difficult or unknown.

KANs use learnable splines for smooth functions. MSN uses power-law bases for singular behavior. Both learn structural parameters; the choice depends on the expected solution structure.

### K.4. Broader Implications

MSN-PINN illustrates a broader principle: Build domain structure directly into the architecture. Rather than using generic approximators and hoping that structure emerges, we design networks whose parameters directly encode physical quantities. This yields four benefits: the learned parameters have clear physical meaning; fewer parameters suffice when the ansatz matches the solution; the embedded physical structure extrapolates beyond the training data; and the learned parameters can be directly compared with theoretical predictions. This philosophy extends beyond power laws. Architectures can learn wavelengths for oscillatory problems, decay rates for exponential phenomena, or ranks for low-rank structures. Parameters that would otherwise be implicit in standard networks become explicit and interpretable.

# L. Closing Remarks

### L.1. Summary of Contributions

We proved when power-law sums with distinct exponents can be uniquely recovered in Theorem 4.4. Our stability analysis in Theorem 4.5 reveals a minimum separation condition analogous to super-resolution limits, explaining why close exponents merge. The constraint-aware convergence theorem in Theorem 4.8 explains why encoding physical requirements improves accuracy by orders of magnitude.

Constraint-aware training in Section 3.4 addresses a key failure mode of naive MSN-PINN: when the PDE provides an insufficient gradient signal, explicit constraint encoding makes the exponents identifiable. Two-timescale optimization in Section 3.5 exploits the asymmetric structure of the MSN loss landscape.

Across six settings, we demonstrated single-exponent recovery with 1–5% error under noise and sparse sampling, the practical limit $\Delta \geq 0.1$ for distinguishing two exponents, physics-informed discovery from PDE constraints alone, and high accuracy on corner singularities with 0.009% error and singular forcing with 0.03% and 0.05% errors.

### L.2. Limitations and Future Directions

Pure MSN fails to represent $x^\alpha \log x$ terms. Extending to log-modified bases is straightforward but requires additional care during optimization. Our 2D wedge experiment succeeded, but three-dimensional corners and edges require further development. A plausible path is to use MSN as a local enrichment or domain-decomposition component near suspected singular sets, analogous in spirit to XFEM enrichment, while retaining standard solvers away from the singular region. MSN currently provides only point estimates. Bayesian extensions could bound uncertainty on discovered exponents. Constraint-aware training assumes a candidate constraint family is available: for wedges, small angle uncertainty can be handled by a validation sweep over $\omega$ and wrong families are detectable by held-out boundary loss (Table 15), but automatically discovering the correct family remains open.

### L.3. Broader Vision

The broader vision is to develop a new approach to scientific machine learning where learned parameters carry physical meaning. Mathematicians and physicists have developed techniques—matched asymptotics, boundary layer theory, and Mellin transforms—to extract scaling structures from differential equations. These methods required expertise and worked only for analytically tractable problems.

MSN offers an alternative: rather than deriving exponents analytically, we learn them from data or physics constraints. Learned exponents carry the same physical meaning as analytical results, but the process is automated and scalable.

Analytical methods remain essential to understand why certain exponents arise and for addressing problems without data. For complex systems that resist analytical treatment—multiphysics problems, irregular geometries, and experimental data—MSN discovers exponents directly. Neural networks should do more than merely fit data. They should reveal the underlying structure.

## M. Extended Introduction

### M.1. The Central Role of Scaling Exponents in Physics

Scaling exponents encode deep physical content and are often the primary scientific goal. When a material fractures, the stress field near the crack tip diverges as $r^{-1/2}$, where $r$ is the distance from the tip (Anderson, 2017). This is not an accident of geometry or material properties; rather, it is a universal consequence of the mathematical structure of elasticity near a singular point. The exponent $-1/2$ encodes physical content: It determines fracture toughness, governs crack propagation speeds, and enables engineers to predict catastrophic failure from local measurements.

The same principle appears throughout physics. In turbulent flows, Kolmogorov (1941) predicts that the energy spectrum scales as $k^{-5/3}$, where $k$ is the wavenumber. Near a second-order phase transition, correlation functions decay as $r^{-\eta}$ with critical exponents $\eta$ that fall into discrete universality classes (Barenblatt, 1996). At the corner of an L-shaped domain, solutions to Laplace's equation behave as $r^{2/3}$, a result first derived by Kondrat'ev (1967). Throughout, the scaling exponent is not merely a fitting parameter; it is a physical observable that encodes symmetries, conservation laws, and the structure of the governing equations.

**Example M.1** (Why Exponents Matter in Practice). Consider a pressure vessel with a small surface crack. The stress intensity factor $K_I$ scales as $K_I \sim \sigma\sqrt{\pi a}$, where $\sigma$ is the applied stress and $a$ is the crack length. The vessel fails when $K_I$ exceeds the material's fracture toughness $K_{IC}$. If an engineer fits experimental stress data with a neural network, they can predict the stress field everywhere. But without knowing the exponent $1/2$, they lack the ability to validate whether their model captures the correct physics, extrapolate to crack lengths outside the training range, connect their measurements to material toughness values, or diagnose whether anomalous predictions indicate model failure or new physics. This exponent is often the primary quantity of physical interest.

### M.2. Neural Networks Approximate but Do Not Explain

Neural networks can approximate any continuous function (Cybenko, 1989), but they prioritize accurate fits over explaining why the function takes a particular form. This poses a problem for scientific applications, which require structural insight in addition to accuracy.

PINNs solve forward and inverse problems involving PDEs by penalizing equation residuals (Raissi et al., 2019; Karniadakis et al., 2021). PINNs encode physical laws directly into the loss function. Yet PINNs still treat the solution as a black box. One can verify that the learned function satisfies the differential equation, but extracting interpretable structure requires fragile, problem-specific post-hoc analysis.

Consider a PINN trained on data near a corner singularity. The network learns an accurate approximation of the solution surface, and the PDE residual converges to zero, confirming physical consistency. However, the singularity exponent is observed only indirectly. Fitting $u(r,\theta) \approx c \cdot r^\alpha \sin(\alpha\theta)$ post-hoc introduces additional errors, and the connection to classical asymptotic theory remains unclear.

Access to exponents makes neural network solutions more useful to scientists. Physical theories predict exponents, not function values at specific points. To validate theory against computation, we must explicitly extract the exponents, which standard architectures typically leave implicit.

### M.3. Why Standard Neural Networks Struggle with Power Laws

Standard neural networks are poorly suited for approximating power-law functions. The difficulty lies in the architecture, not just training. A standard MLP with ReLU activations represents functions as

$$f(x) = \sum_{j=1}^{N} c_j \max(0, w_j x + b_j). \tag{45}$$

Each ReLU unit contributes a piecewise-linear hinge function. To approximate $x^\alpha$ for $\alpha \in (0, 1)$, the network must compose many such hinges to create curvature. As shown by N'guessan (2025), achieving $\epsilon$-accuracy for $x^{0.5}$ on $[0, 1]$ requires $N = \Omega(\epsilon^{-2})$ ReLU units.

The situation is worse than this count suggests. Even with sufficient units, the network encodes $x^{0.5}$ implicitly rather than revealing it as the natural form. The exponent 0.5 spreads across millions of weight parameters, defying interpretation. Fitting $c \cdot x^\alpha$ to the learned function reintroduces error and leaves the true value uncertain.

Smooth activations like $\tanh$ or sigmoid face different but severe challenges. Because these functions and their derivatives are bounded, representing the unbounded derivatives of $x^\alpha$ near $x = 0$ requires cancellation among many units. Consequently, the resulting networks tend to be brittle and difficult to train.

### M.4. Significance and Outlook

MSN connects two intellectual traditions that have developed largely independently: Asymptotic analysis, the classical approach to singular problems, reveals scaling structure through matched expansions, boundary layer theory, and Mellin transforms. These methods require human expertise, work only for analytically tractable problems, and involve guessing solution forms.

Neural networks approximate complex, high-dimensional problems without requiring analytical insight—but they hide structure. The learned solution remains disconnected from physical theory.

MSN makes exponents explicit physical parameters, like asymptotic analysis does, while retaining the flexibility of neural networks—no analytical solution is required. The discovered exponents validate theory against analytical predictions, suggest new physics when they reveal unexpected scaling regimes, diagnose models when exponent drift indicates misspecification, and enable extrapolation because the power-law structure extends naturally beyond training data.

