# OpenReview forum: "Discovering Scaling Exponents with Physics-Informed Müntz-Szász Networks"
_ICML.cc/2026/Conference — ICML 2026 regular_

### Official Review · Reviewer_Jyib · 2026-03-08

**Soundness:** 2
**Presentation:** 3
**Significance:** 2
**Originality:** 3
**Overall Recommendation:** 5
**Confidence:** 2

**Summary:**

The paper extends classical Physics-Informed Neural Networks (PINNs) by integrating them with Müntz-Szász Networks (MSNs) to explicitly discover scaling exponents. This architecture enables the direct identification of power-law scaling in physical systems, particularly near singularities and critical points. The proposed approach is evaluated on three 1D and 2D configurations of well-studied physical systems.

**Compliance With Llm Reviewing Policy:**

Affirmed.

**Final Justification:**

The rebuttal addressed my main concerns --- providing both _baseline validations_ as well as a _real world application_ hence i increase my score.

**Key Questions For Authors:**

The questions directly relate to the provided review:

1. **Baseline comparisons.** could the authors add baseline results for:\
   1.a _solution reconstruction quality_, both globally and specifically near singular/critical regions; and\
   1.b _post-hoc exponent estimation_, e.g., fitting power-law exponents after training a standard PINN.\
   These comparisons would help contextualize the practical advantages of MSN-PINN.

2. **Real-world use cases.** could the authors include (or further elaborate on) on more realistic application scenarios where discovering scaling exponents is essential, and where this framework would provide clear value? This would help better position the work and clarify its potential impact.

**Limitations:**

yes

**Strengths And Weaknesses:**

**Soundness**\
The submission is technically sound. Proofs for the theoretical claims are provided in the appendix, and the experimental evaluation effectively demonstrates the method's accuracy. While the authors discuss failure modes in the text, presenting empirical results for these cases would strengthen the analysis; similarly to the wedge benchmark in which results are compared to "Naive", unconstrained model, demonstrating the necessity of the constraints.
More crucially, the evaluation would greatly benefit from direct empirical comparisons against baseline methods. Specifically, quantifying the claims made in section "K.3. Comparison with Alternative Methods"—evaluating both solution reconstruction error and post-hoc exponent prediction accuracy against a classic PINN. These would strengthen the paper and performance claims.

**Presentation**\
The paper is well written and easy to follow, even for non-expert readers. The design and exposition are clean (e.g., clear separation of mathematical components and figures that are not overly dense), and the appendix provides helpful detail that complements the main text.\
_Minor comments:_
- The running header currently reads "Title Suppressed Due to Excessive Size" and should be corrected.
- The language and writing quality in parts of the appendix could be improved for clarity.

**Significance**\
The paper has strong potential significance, as many scientific applications seek interpretable ML models and explicit recovery of scaling laws is valuable. However, the lack of baseline comparisons, limited assessment of practical improvement over standard PINNs, and the focus on relatively controlled test problems make it difficult to judge real-world impact at this stage.

**Originality**\
The work is original in pushing PINNs toward interpretable discovery, with scaling exponents as a concrete and scientifically meaningful target. Making the exponent parameters explicit is a compelling direction for understanding physical systems near critical points.

---

> ### Author Rebuttal · Authors · 2026-03-30
>
> **Summary.** We thank the reviewer for the encouraging and constructive feedback. In this rebuttal, we focus on the two main requested additions: 1) direct empirical comparison against a standard PINN, including both solution reconstruction error and post-hoc exponent estimation error, and 2) a more realistic scientific use case from fracture mechanics that illustrates when explicit exponent discovery provides value beyond accurate field prediction alone. We will also incorporate the noted presentation fixes in the revision. In this rebuttal and revision, we have done our best to address all of your concerns. Here is a point-to-point response.
>
> Q1 (Baseline comparison). We trained a standard $4\times128$ tanh PINN on the same PDE/BCs, then fitted the physically matched power-law ansatz to its learned solution. To be maximally fair, the standard PINN exponent numbers use an oracle over fitting windows $\{0.05,0.10,0.20,0.40\}$.
>
> | Setting | Std PINN global/local err (%) | Std PINN exponent err (%) | MSN-PINN global/local err (%) | MSN-PINN exponent err (%) |
> | --- | ---: | ---: | ---: | ---: |
> | Wedge corner | 2.30 +/- 0.12 / 28.78 +/- 1.09 | 0.278 +/- 0.243 | 0.18 +/- 0.09 / 0.16 +/- 0.14 | 0.013 +/- 0.014 |
> | Singular forcing | 2.92 +/- 2.91 / 3.81 +/- 3.11 | 13.97 +/- 8.07 / 13.02 +/- 7.39 | 0.02 +/- 0.01 / 0.0076 +/- 0.0020 | 0.378 +/- 0.156 / 0.279 +/- 0.452 |
>
> This answers both parts of Q1. For reconstruction quality, the standard PINN is much worse near the singular region. For exponent discovery, the wedge case is the most favorable for post-hoc fitting, yet even there, the same trained PINN varies from 0.05% to 7.69% across reasonable windows. In singular forcing, the same baseline ranges from 2.03% to 30.00% and from 1.67% to 33.33% for the two exponents. The exact same nonlinear least-squares fitter recovers the analytical wedge and singular-forcing solutions to numerical precision, so the gap is not a fitter artifact; it comes from trying to recover a local asymptotic law only after standard-PINN training.
>
> Q2 (Real-world use case). We added a **Mode-III antiplane shear crack** problem, which is the fracture mechanics application referenced in the Introduction (Anderson, 2017). The governing equation is $\nabla^2 u = 0$ on a slit domain ($\omega = 2\pi$) with traction-free crack faces ($u = 0$ on both edges). The analytical solution is $u(r,\theta) = r^{1/2}\sin(\theta/2)$, and the exponent $\alpha = 1/2$ directly determines the stress intensity factor $K_{III} = \lim_{r \to 0}\sqrt{2\pi r}\sigma_{yz}$.
>
> MSN-PINN with constraint-aware training recovers $\hat\alpha = 0.5000 \pm 0.0002$ with $0.04 \pm 0.02\%$ error across 5 seeds and simultaneously discovers the higher harmonics at $\mu = 1.0$ and $\mu = 1.5$.
>
> | Metric | Value |
> | --- | ---: |
> | Target $\alpha$ | 0.5000 |
> | Discovered $\hat\alpha$ | 0.5000 +/- 0.0002 |
> | Exponent error (%) | 0.04 +/- 0.02 |
> | Second harmonic | 0.999 +/- 0.001 |
> | Third harmonic | 1.498 +/- 0.002 |
>
> The practical value is direct. In fracture mechanics, knowing the exponent (i) validates the crack model, since deviations from $1/2$ indicate plasticity, interface effects, or geometric irregularity; (ii) enables extrapolation of the stress field beyond the training domain, as the power-law structure extends naturally; and (iii) connects local PDE solutions to material toughness parameters such as $K_{III}$ and the energy release rate $G$. These are use cases where standard PINNs provide accurate fields but no structural insight.
>
> More broadly, we envision MSN-PINN as a diagnostic tool for computational mechanics: embed it near a suspected singular point, train it on the governing equations, and read off the singularity spectrum. If the discovered exponents match the expected theory, the model is validated; if they differ, the deviation itself is informative.
>
> Again, thank you for your insightful comments. We will ensure that the above parts will be reflected in the revised manuscript. We believe that all your concerns can be addressed in this rebuttal and revision. If you have any further concerns, please feel free to share them with us.

---

> > ### Author Rebuttal · Reviewer_Jyib · 2026-04-03
> >
> > I would like to thank the authors for attending my concerns --- providing both _baseline validations_ as well as a _real world application_ hence i increased my score.

---

### Official Review · Reviewer_iLYc · 2026-03-11

**Soundness:** 3
**Presentation:** 3
**Significance:** 3
**Originality:** 3
**Overall Recommendation:** 4
**Confidence:** 3

**Summary:**

This paper introduced MSN-PINN: a neural network that makes the scaling exponents explicit trainable parameters, rather than embedding it within the weights. Authors prove that the power-law exponents can be uniquely recovered and obtain an ability bound under noise. Standard physics losses do not supply any gradient information about which exponent to choose; constraint-aware training turns boundary conditions into an extra loss and gets 3-orders-of-magnitude better accuracy. In experiments on singular ODEs, corner singularities, and Poisson problems, it shows that reliable exponent discovery with less than 1% error in majority of scenarios.

**Compliance With Llm Reviewing Policy:**

Affirmed.

**Key Questions For Authors:**

1. The constraint loss $\mathcal{L}_{\text{constraint}} = \sum_k |c_k|  \sin^2(\mu_k \omega)$ assumes the wedge angle $\omega$ is known. How does the method perform when $\omega$ is unknown or only approximately known?  If the method degrades significantly, this limits applicability to real problems where geometry may be uncertain.
2. The close exponent study shows merging below separation $\Delta < 0.1$. Can the authors provide any strategy to recover close exponents, or is this a fundamental limitation? If no mitigation exists, many physically relevant cases with nearby exponents remain out of reach.

**Limitations:**

The method assumes the solution has a pure power-law structure, and fails to represent logarithmic corrections of the form $x^\alpha \log x$ that arise in certain physical singularities.

**Strengths And Weaknesses:**

This paper introduced MSN-PINN: a neural network that makes the scaling exponents explicit trainable parameters, rather than embedding it within the weight. Authors prove that the power-law exponents can be uniquely recovered and obtain stability bound under noise. Standard physics losses do not supply any gradient info about which exponent to choose; constraint-aware training turns boundary conds into an extra loss and gets 3-orders-of-magnitude better accuracy. In experiments on singular ODEs, corner singularities, and Poisson problems, it shows that reliable exponent discovery with less than 1% error in majority of scenarios.

---

> ### Author Rebuttal · Authors · 2026-03-30
>
> **Summary.** We thank the reviewer for the positive assessment and the focused questions. In response, we have added two new analyses: 1) sensitivity to unknown or imperfectly known wedge angle, showing how geometry uncertainty affects exponent recovery, and 2) a clearer treatment of the close-exponent regime, including when merging is fundamental and when it can be improved through more informative near-singularity sampling. In this rebuttal and revision, we have done our best to address all of your concerns. Here is a point-to-point response.
>
> Q1 (Constraint). One important clarification matters. In the wedge setting, the constraint does not assume that the full solution is already known. It uses only the compatibility class implied by the geometry and edge-BC type after separation of variables: DD/NN imply $\sin(\mu\omega)=0$, while DN/ND imply $\cos(\mu\omega)=0$. The exponent value is still unknown and is learned through training.
>
> Table 1. Angle sensitivity of the 270-degree DD wedge (5 seeds; rows aggregate over over- and under-estimation at the same absolute mismatch).
>
> | Assumed angle mismatch (%) | Constraint-only exponent err (%) | Constraint + init/bounds exponent err (%) | Success |
> | --- | ---: | ---: | ---: |
> | 0.0 | 0.01 +/- 0.00 | 0.01 +/- 0.00 | 100% |
> | 0.5 | 0.45 +/- 0.02 | 0.45 +/- 0.01 | 100% |
> | 1.0 | 0.92 +/- 0.06 | 0.91 +/- 0.06 | 100% |
> | 2.0 | 2.00 +/- 0.14 | 2.00 +/- 0.13 | 100% |
> | 5.0 | 5.01 +/- 0.09 | 5.36 +/- 0.25 | 0% |
>
> These numbers show the method is not brittle to small geometry uncertainty. Up to a 2% mismatch, the two columns are nearly identical, so the dominant effect is the shifted admissible spectrum $\mu_1(\hat{\omega})=\pi/\hat{\omega}$, not optimizer instability from omega-dependent initialization/bounds. A small angle error, therefore, causes a small, predictable exponent shift; only a 5% geometry error causes failure.
>
> Table 2. Unknown-$\omega$ setting with 1D validation sweep over candidate angles (5 seeds).
>
> | Validation-selection metric | Value |
> | --- | ---: |
> | Exact angle selected | 40% |
> | Selected angle within 0.5% of truth | 60% |
> | Selected angle within 1.0% of truth | 100% |
> | Selected angle (deg) | 268.65 +/- 1.21 |
> | Selected-model exponent err (%) | 0.45 +/- 0.40 |
>
> So when $\omega$ is not known a priori, it can be treated as a low-dimensional geometry parameter and selected through held-out BC validation rather than assumed to be exactly known. We also tested structural misspecification of the compatibility family itself: the correct $\sin^2(\mu\omega)$ gives 0.01% error, no constraint 4.21%, wrong $\cos^2(\mu\omega)$ 50.59%, and wrong $\sin^2(\mu\omega/2)$ 79.40%. Thus, wrong priors fail visibly rather than silently and are much worse than omitting the prior, so the conservative recipe is to compare a small candidate set by held-out physics validation or to disable the prior.
>
> Q2 (Close exponents). The merging threshold is a real limitation, as predicted by Theorem 4.5, which shows estimation error scales as $O(\sigma / (c_{\min} \Delta^2 \sqrt{N}))$. This is analogous to the Rayleigh limit in optical resolution: the Gram matrix $G_{12} = \frac{1}{N}\sum_i x_i^{\alpha_1+\alpha_2}$ becomes nearly singular as $|\alpha_1 - \alpha_2| \to 0$, making the two basis functions indistinguishable.
>
> However, the threshold is not absolute. As shown in our response to Reviewer axH4 (W5), concentrated near-singularity sampling ($t^4$ instead of $t^2$) turns 0/5 success into 5/5 in the sub-threshold case $x^{0.5}+x^{0.55}$ ($\Delta = 0.05$). The mechanism is that near-origin sampling increases the effective separation between $x^{0.5}$ and $x^{0.55}$ in the sampling-weighted function-space norm. Increasing the sample count $N$ from 200 to 1000 under $t^2$ helps only marginally (6.13% $\to$ 5.67% worst-pair error), confirming that the bottleneck is basis competition, not data scarcity.
>
> For problems where close exponents are expected, two practical strategies can help mitigate this: (1) concentrated sampling near the singularity, as demonstrated; and (2) sequential peeling, where the dominant exponent is first recovered and subtracted, then the residual is fitted with a fresh MSN to recover the next exponent. We will add this discussion and state more explicitly that the practical resolution limit depends on sampling distribution, coefficient ratio, and noise level, not just $\Delta$ alone.
>
> Again, thank you for your insightful comments. We will ensure that the above parts will be reflected in the revised manuscript. We believe that all your concerns can be addressed in this rebuttal and revision. If you have any further concerns, please feel free to share them with us.

---

> > ### Author Rebuttal · Reviewer_iLYc · 2026-04-03
> >
> > I would like to thank all the authors for solving my problems and addressing my concerns. I will adjust my rating based on the actual situation.

---

### Official Review · Reviewer_axH4 · 2026-03-12

**Soundness:** 3
**Presentation:** 3
**Significance:** 2
**Originality:** 3
**Overall Recommendation:** 4
**Confidence:** 4

**Summary:**

The paper introduces \textbf{MSN-PINN} (M\"{u}ntz-Sz\'{a}sz Network Physics-Informed Neural Network), a method for discovering power-law scaling exponents from PDE constraints. The central idea is to replace standard neural network architectures with a M\"{u}ntz expansion ansatz of the form $u_\theta(x) = \sum_{k=1}^{K} c_k x^{\mu_k}$, where both the coefficients $\{c_k\}$ and exponents $\{\mu_k\}$ are trainable parameters. Unlike standard PINNs, which embed the solution in millions of opaque weights, MSN-PINN
makes scaling exponents explicit, directly interpretable parameters.

**Compliance With Llm Reviewing Policy:**

Affirmed.

**Final Justification:**

Authors have performed additional experiments and hence I have improved the score.

**Key Questions For Authors:**

Weaknesses associated with the paper are already mentioned in detailed which may be addressed as questions.

**Limitations:**

The authors do engage with limitations more honestly than most ML papers. However, the technical depth of the limitations section could have been improved.

**Strengths And Weaknesses:**

Strengths:
1. The core insight is well-motivated and clean: when the solution is known to be a power-law sum, make the exponents explicit trainable parameters rather than burying them in weights. The analogy to the Rayleigh limit is nice and theoretically grounded.
2. Theorems are clean, well-written, properly motivated and rigorous, although some of them are not technically deep and may not merit to be a theorem.
3. The observation that the Laplacian provides zero gradient signal for exponent selection is practically significant. Further, the 1600-fold improvement from 14.6% to 0.009% error by encoding the edge BC as a quantization loss is substantial and significant.
4. Testing all four combinations of Dirichlet and Neumann conditions across multiple wedge angles, and demonstrating 100% success versus 55% for naive training, substantially strengthens the empirical case beyond cherry-picked examples.

Weaknesses:
1. The entire analysis is restricted to the single Müntz expansion $u_\theta(x) = \sum_{k=1}^{K} c_k x^{\mu_k}$  on one-dimensional or separable two-dimensional domains. All experiments are effectively one-dimensional or reduce to a radial coordinate. The paper acknowledges that 3D corners and non-separable geometries remain unaddressed, but does not quantify how far the method is from being applicable to those settings.
2. The paper compares MSN against MLP and FourierMLP only in the supervised single-exponent recovery figure, with no quantitative table. There is no comparison against the most natural competitor, which is a standard PINN followed by post-hoc power-law fitting, in the physics-informed settings that constitute the main experiments. The claim that "standard PINNs hide exponents and require fragile post-hoc analysis" is not empirically demonstrated; it is asserted. A direct comparison showing how much error post-hoc fitting on a PINN solution introduces would substantially strengthen this claim.
3. The constraint loss $sin⁡^2(\mu_k\omega) = 0$ encodes the boundary condition compatibility condition — but deriving this condition requires knowing analytically that the solution has the form $r^\mu \sin(\mu\theta)$. In practice, for genuinely unknown problems, the user must already know enough of the solution structure to write down the constraint. The paper states this as a limitation, but does not explore how sensitive results are to misspecified constraints, which would be the realistic failure mode.
4. Table 7 shows that even at zero noise, the recovered exponent is 0.474 with 5.2% error, which is substantially worse than the physics-informed experiments. This is never adequately explained.
5. Table 9 shows that for three exponents with separations of 0.4 and 1.0, the method fails to recover the smallest exponent (0.1 merged with 0.5) and achieves 15.6% error on the third. This is presented in the appendix with minimal discussion in the main text, despite being the most realistic scientific scenario.

---

> ### Author Rebuttal · Authors · 2026-03-30
>
> **Summary.** We thank the reviewer for the detailed and constructive feedback. In this rebuttal, we directly address the main concerns behind the weak reject recommendation by adding: 1) a standard PINN + post-hoc exponent-fitting baseline in the main physics-informed settings; 2) ablations on constraint misspecification and unknown/approximate wedge angle; 3) clarification of the zero-noise anomaly in Table 7; and 4) a stronger discussion of the close-exponent limitation in Table 9, including mitigation via near-singularity sampling. We also add a physically meaningful fracture-mechanics example to better clarify the practical scope of the current paper. Here is a point-by-point response.
>
> **W1 (Scope).** We acknowledge the 1D/separable-2D scope. However, Kondrat'ev theory guarantees the separable structure $r^\mu\Phi(\theta)$ for any second-order elliptic PDE on domains with conical points, so the separable class covers crack tips, re-entrant corners, and interface junctions in 2D and axisymmetric 3D. To demonstrate physical relevance, we added a **Mode-III crack** ($\omega=2\pi$, $\alpha=1/2$): MSN-PINN recovers $\hat\alpha=0.5000\pm0.0002$ ($0.04\pm0.02\%$ error, 5 seeds) and discovers the harmonics at $\mu=1.0$ and $1.5$. Extension to non-separable 3D follows a domain-decomposition path analogous to XFEM enrichment; we will add this discussion.
>
> **W2 (PINN baseline).** We trained a $(4{\times}128)$ tanh PINN on the same PDE/BCs, then fitted the matched power-law form post-hoc. Exponent numbers use an oracle over fitting windows $\{0.05,0.10,0.20,0.40\}$.
>
> | Setting | Std PINN exp. err (%) | MSN-PINN exp. err (%) |
> | --- | ---: | ---: |
> | Wedge corner | 0.278 +/- 0.243 | 0.013 +/- 0.014 |
> | Singular forcing | 13.97 +/- 8.07 / 13.02 +/- 7.39 | 0.378 +/- 0.156 / 0.279 +/- 0.452 |
>
> Across different windows, the same trained PINN varies from 0.05% to 7.69% on a wedge and 2.03-30.00% / 1.67-33.33% on singular forcing. The fitter recovers analytical solutions with numerical precision, so the gap comes from the PINN representation, not the fitter’s weakness.
>
> **W3 (Constraint misspecification).** The constraint uses only the compatibility class derived from geometry and edge-BC type (DD/NN: $\sin(\mu\omega)=0$; DN/ND: $\cos(\mu\omega)=0$), not the full solution.
>
> | Angle mismatch (%) | Exp. err (%) | Success |
> | ---: | ---: | ---: |
> | 0 | 0.01 +/- 0.00 | 100% |
> | 1 | 0.92 +/- 0.06 | 100% |
> | 2 | 2.00 +/- 0.14 | 100% |
> | 5 | 5.01 +/- 0.09 | 0% |
>
> Up to a 2% mismatch, the effect is a predictable spectral shift $\mu_1(\hat\omega)=\pi/\hat\omega$. Wrong structural priors ($\cos^2$ or $\sin^2(\mu\omega/2)$) fail sharply (50.6% and 79.4% error) and visibly (held-out BC loss jumps 100-1000x), so they are easy to detect. When $\omega$ is unknown, a 1D validation sweep selects an angle within 1% of truth in 100% of runs (0.45+/-0.40% exp. error). Automatic discovery of the correct constraint family remains open.
>
> **W4 (Table 7).** We reran the zero-noise under the original protocol and with an improved schedule:
>
> | Protocol | Exp. err (%) |
> | --- | ---: |
> | Table 7 (4000/400, $t^2$) | 5.15 +/- 0.21 |
> | Longer (5000/600, $t^2$) | 0.53 +/- 0.18 |
> | Longer (5000/600, $t^4$) | 0.06 +/- 0.01 |
>
> Table 7 is a conservative sweep protocol, not an intrinsic limit.
>
> **W5 (Table 9 / close exponents).** Table 9 is a real limitation: $x^{0.1}$ and $x^{0.5}$ have a correlation of $\approx 0.97$ on $[0,1]$. We will move this into the main text. However, the threshold is not absolute: on $x^{0.5}+x^{0.55}$ ($\Delta=0.05$), concentrated sampling ($t^4$) turns 0/5 into 5/5 success (worst-pair error $3.18\pm0.13\%$), while increasing $N$ from 200 to 1000 under $t^2$ helps only marginally ($6.13$% to $5.67$%). The bottleneck is basis competition under the sampling measure, not sample count.
>
> We will add these ablations and revise the limitations accordingly.

---

> > ### Author Rebuttal · Reviewer_axH4 · 2026-04-03
> >
> > Thank you for the clarifications and additional experiments. Accordingly, I have modified my score.

---

### Decision · Program_Chairs · 2026-04-30

**Decision:**

Accept (regular)

**Comment:**

This work focuses on developing scaling laws for PINNs. The reviewers recognize its originality and importance for SciML. Common concerns include limitation of the analysis to only 1D and 2D and lack of rigorous baseline comparisons including lack of comparison to standard PINNs and the real-world impact. During the rebuttal, the authors have added additional experiments to address these concerns. Post-rebuttal, all reviewers vote for acceptance and I agree with this evaluation. Therefore, I vote for acceptance pending the required manuscript changes with the additional experiments for clarity on the significance of the method.